# Adaptive Ensemble Q-learning: Minimizing Estimation Bias via Error Feedback

**Hang Wang**
Arizona State University
Tempe, Arizona, USA
hwang442@asu.edu

**Sen Lin**
Arizona State University
Tempe, Arizona, USA
slin70@asu.edu

**Junshan Zhang**
Arizona State University
Tempe, Arizona, USA
Junshan.Zhang@asu.edu

## Abstract

The ensemble method is a promising way to mitigate the overestimation issue in Q-learning, where multiple function approximators are used to estimate the action values. It is known that the estimation bias hinges heavily on the ensemble size (i.e., the number of Q-function approximators used in the target), and that determining the 'right' ensemble size is highly nontrivial, because of the time-varying nature of the function approximation errors during the learning process. To tackle this challenge, we first derive an upper bound and a lower bound on the estimation bias, based on which the ensemble size is adapted to drive the bias to be nearly zero, thereby coping with the impact of the time-varying approximation errors accordingly. Motivated by the theoretic findings, we advocate that the ensemble method can be combined with Model Identification Adaptive Control (MIAC) for effective ensemble size adaptation. Specifically, we devise Adaptive Ensemble Q-learning (AdaEQ), a generalized ensemble method with two key steps: (a) approximation error characterization which serves as the feedback for flexibly controlling the ensemble size, and (b) ensemble size adaptation tailored towards minimizing the estimation bias. Extensive experiments are carried out to show that AdaEQ can improve the learning performance than the existing methods for the MuJoCo benchmark.

## 1 Introduction

Thanks to recent advances in function approximation methods using deep neural networks [20], Q-learning [35] has been widely used to solve reinforcement learning (RL) problems in a variety of applications, e.g., robotic control [23, 13], path planning [15, 24] and production scheduling [34, 21]. Despite the great success, it is well recognized that Q-learning may suffer from the notorious overestimation bias [29, 33, 32, 10, 37], which would significantly impede the learning efficiency. Recent work [9, 11] indicates that this problem also persists in the actor-critic setting. To address this issue, the ensemble method [16, 1, 26, 7] has emerged as a promising solution in which multiple Q-function approximators are used to get better estimation of the action values. Needless to say, the ensemble size, i.e., the number of Q-function approximators used in the target, has intrinsic impact on Q-learning. Notably, it is shown in [6, 17] that while a large ensemble size could completely remove the overestimation bias, it may go to the other extreme and result in underestimation bias and unstable training, which is clearly not desirable. Therefore, instead of simply increasing the ensemble

size to mitigate the overestimation issue, a fundamental question to ask is:" *Is it possible to determine the right ensemble size on the fly so as to minimize the estimation bias*?"

Some existing ensemble methods [2, 19, 17] adopt a trial-and-error strategy to search for the ensemble size, which would be time-consuming and require a lot of human engineering for different RL tasks. The approximation error of the Q-function during the learning process plays a nontrivial role in the selection of the ensemble size, since it directly impacts the Q-target estimation accuracy. This however remains not well understood. In particular, the fact that the approximation error is time-varying, due to the iterative nature of Q-learning [36, 5], gives rise to the question that whether a *fixed* ensemble size should be used in the learning process. To answer this question, we show in Section 2.2 that using a fixed ensemble size is likely to lead to either overestimation or underestimation bias, and the bias may shift between overestimation and underestimation because of the time-varying approximation error, calling for an adaptive ensemble size so as to drive the bias close to zero based on the underlying learning dynamics.

Thus motivated, in this work we study effective ensemble size adaptation to minimize the estimation bias that hinges heavily on the time-varying approximation errors during the learning process. To this end, we first characterize the relationship among the ensemble size, the function approximation errors, and the estimation bias, by deriving an upper bound and a lower bound on the estimation bias. Our findings reveal that the ensemble size should be selected adaptively in a way to cope with the impact of the time-varying approximation errors. Building upon the

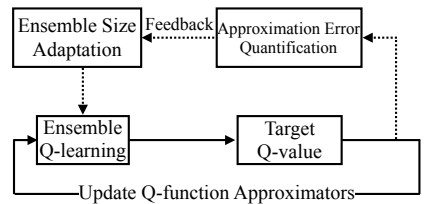

Figure 1: A sketch of the adaptive ensemble Q-learning (AdaEQ).

theoretic results, we cast the estimation bias minimization as an adaptive control problem where the approximation error during the learning process is treated as the control object, and the ensemble size is adapted based on the feedback of the control output, i.e., the value of the approximation error from the last iteration. The key idea in this approach is inspired from the classic Model Identification Adaptive Control (MIAC) framework [3, 25], where at each step the current system identification of the control object is fed back to adjust the controller, and consequently a new control signal is devised following the updated control law.

One main contribution of this work lies in the development of AdaEQ, a generalized ensemble method for the ensemble size adaptation, aiming to minimize the estimation bias during the learning process. Specifically, the approximation error in each iteration is quantified by comparing the difference between the Q-estimates and the Monte Carlo return using the current learned policy over a testing trajectory [29, 17]. Inspired by MIAC, the approximation error serves as the feedback to adapt the ensemble size. Besides, we introduce a 'tolerance' parameter in the adaptation mechanism to balance the control tendency towards positive or negative bias during the learning process. In this way, AdaEQ can encompass other existing ensemble methods as special cases, including Maxmin [17], by properly setting this hyperparameter. A salient feature of the feedback-adaptation mechanism is that it can be used effectively in conjunction with both standard Q-learning [22] and actor-critic methods [28, 11]. Experimental results on the continuous-control MuJoCo benchmark [30] show that AdaEQ is robust to the initial ensemble size in different environments, and achieves higher average return, thanks to keeping the estimation bias close to zero, when compared to the state-of-the-art ensemble methods such as REDQ [6] and Average-DQN [2].

**Related Work.** Bias-corrected Q-learning [18] introduces the bias correction term to reduce the overestimation bias. Double Q-learning is proposed in [12, 33] to address the overestimation issue in vanilla Q-learning, by leveraging two independent Q-function approximators to estimate the maximum Q-function value in the target. S-DQN and S-DDQN use the softmax operator instead of the max operator to further reduce the overestimation bias [27]. Self-correcting Q-learning aims to balance the underestimation in double Q-learning and overestimation in classic Q learning by introducing a new self-correcting estimator [38]. Weighted Q-learning proposes a new estimator based on the weighted average of the sample means, and conducts the empirical analysis in the discrete action space [8]. Weighted Double Q-learning [37] uses the Q-approximator together with the double Q-approximator to balance the overestimation and underestimation bias. Nevertheless, acquiring independent approximators is often intractable for large-scale tasks. To resolve this issue, the Twin-Delayed Deep Deterministic policy gradient algorithm (TD3) [9] and Soft Actor-Critic

(SAC) [11] have been devised to take the minimum over two approximators in the target network. Along a different avenue, the ensemble-based methods generalize double Q-learning to correct the overestimation bias by increasing the number of Q-function approximators. Particularly, Average-DQN [2] takes the average of multiple approximators in the target to reduce the overestimation error, and Random Ensemble Mixture (REM) [1] estimates the target value using the random convex combination of the approximators. It is worth noting that both Average-DQN and REM cannot completely eliminate the overestimation bias. Most recently, Maxmin Q-learning [17] defines a proxy Q-function by choosing the minimum Q-value for each action among all approximators. Similar to Maxmin, Random Ensembled Q-learning (REDQ) [6] formulates the proxy Q-function by choosing only a subset of the ensemble. Nevertheless, both Maxmin and REDQ use a fixed ensemble size. In this study, we introduce an adaptation mechanism for the ensemble size to drive the estimation bias to be close to zero, thereby mitigating the possible overestimation and underestimation issues.

## 2    Impact of Ensemble Size on Estimation Bias

### 2.1    Ensemble Q-learning

As is standard, we consider a Markov decision process (MDP) defined by the tuple $\langle \mathcal{S}, \mathcal{A}, P, r, \gamma \rangle$, where $\mathcal{S}$ and $\mathcal{A}$ denote the state space and the action space, respectively. $P(s'|s, a) : \mathcal{S} \times \mathcal{A} \times \mathcal{S} \rightarrow [0, 1]$ denotes the probability transition function from current state $s$ to the next state $s'$ by taking action $a \in \mathcal{A}$, and $r(s, a) : \mathcal{S} \times \mathcal{A} \rightarrow \mathbb{R}$ is the corresponding reward. $\gamma \in (0, 1]$ is the discount factor. At each step $t$, the agent observes the state $s_t$, takes an action $a_t$ following a policy $\pi : \mathcal{S} \rightarrow \mathcal{A}$, receives the reward $r_t$, and evolves to a new state $s_{t+1}$. The objective is to find an optimal policy $\pi^*$ to maximize the discounted return $R = \sum_{t=0}^{\infty} \gamma^t r_t$.

By definition, Q-function is the expected return when choosing action $a$ in state $s$ and following with the policy $\pi$: $Q^\pi = \mathbf{E}[\sum_{t=0}^{\infty} \gamma^t r_t(s_t, a_t)|s_0 = s, a_0 = a]$. Q-learning is an off-policy value-based method that aims at learning the optimal Q-function $Q^* : \mathcal{S} \times \mathcal{A} \rightarrow \mathbb{R}$, where the optimal Q-function is a fixed point of the Bellman optimality equation [4]:

$$\mathcal{T}Q^*(s, a) = r(s, a) + \gamma \mathbf{E}_{s' \sim P(s'|s,a)} \left[\max_{a' \in \mathcal{A}} Q^*(s', a')\right]. \qquad (1)$$

Given a transition sample $(s, a, r, s')$, the Bellman operator can be employed to update the Q-function as follows:

$$Q(s, a) \leftarrow (1 - \alpha)Q(s, a) + \alpha y, \quad y := r + \gamma \max_{a' \in \mathcal{A}} Q(s', a'). \qquad (2)$$

where $\alpha$ is the step size and $y$ is the target. Under some conditions, Q-learning can converge to the optimal fixed-point solution asymptotically [31]. In deep Q-learning, the Q-function is approximated by a neural network, and it has been shown [33] that the approximation error, amplified by the $\max$ operator in the target, results in the overestimation phenomena. One promising approach to address this issue is the ensemble Q-learning method, which is the main subject of this study.

**The Ensemble Method.** Specifically, the ensemble method maintains $N$ separate approximators $Q^1, Q^2, \cdots, Q^N$ of the Q-function, based on which a subset of these approximators is used to devise a proxy Q-function. For example, in Average-DQN [2], the proxy Q-function is obtained by computing the average value over all $N$ approximators to reduce the overestimation bias:

$$Q^{\text{ave}}(\cdot) = \tfrac{1}{N} \sum_{i=1}^{N} Q^i(\cdot).$$

However, the average operation cannot completely eliminate the overestimation bias, since the average of the overestimation bias is still positive. To tackle this challenge, Maxmin [17] and REDQ [6] take the 'min' operation over a subset $\mathcal{M}$ ( size $M$) of the ensemble:

$$Q^{\text{proxy}}(\cdot) = \min_{i \in \mathcal{M}} Q^i(\cdot). \qquad (3)$$

The target value in the ensemble-based Q-learning is then computed as $y = r + \max_{a' \in \mathcal{A}} Q^{\text{proxy}}$. It is worth noting that in the existing studies, the in-target ensemble size $M$, pre-determined for a given environment, remain fixed in the learning process.

### 2.2    An Illustrative Example

It is known that the determination of the optimal ensemble size is highly nontrivial, and a poor choice of the ensemble size would degrade the performance of ensemble Q-learning significantly [17]. As mentioned earlier, it is unclear a priori if a fixed ensemble size should be used in the learning process.

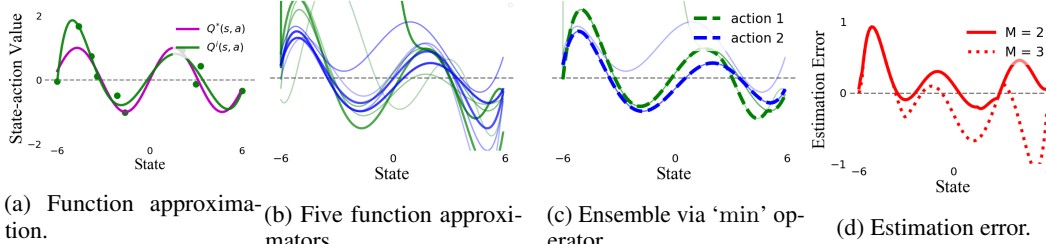

(a) Function approximation.

(b) Five function approximators.

(c) Ensemble via 'min' operator.

(d) Estimation error.

Figure 2: Illustration of estimation bias in the ensemble method. (a) Each approximator is fitted to the noisy values (green dots) at the sampled states independently. (b) Five Q-function approximators are obtained for both actions (green lines and blue lines). (c) Apply the $\min$ operator over $M$ ($M = 3$) randomly selected approximators to obtain a proxy approximator for each action. (d) The estimation error is obtained by comparing the underlying true value (purple line in (a)) and the target value using the proxy approximator.

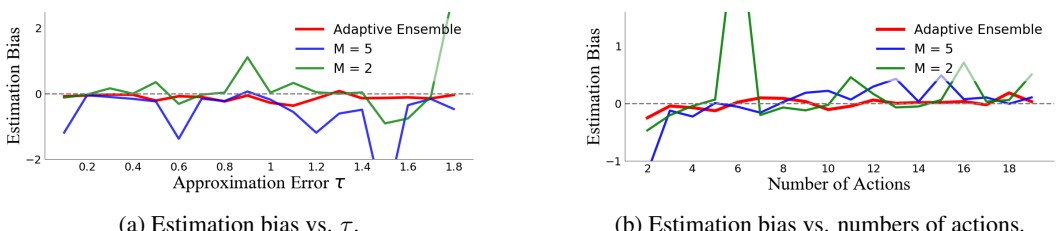

(a) Estimation bias vs. $\tau$.

(b) Estimation bias vs. numbers of actions.

Figure 3: Illustration of overestimation and underestimation phenomena for different ensemble sizes.

In what follows, we use an example to illustrate the potential pitfalls in the ensemble methods by examining the sensitivity of the estimation bias to the ensemble size [6, 17].

Along the same line as in [33], we consider an example with a real-valued continuous state space. In this example, there are two discrete actions available at each state and the optimal action values depend only on the state, i.e., in each state both actions result in the same optimal value $Q^*(s, \cdot)$, which is assumed to be $Q^*(s, \cdot) = \sin(s)$. Figure 2 demonstrates how the ensemble method is carried out in four stages:

(I) For each Q-function approximator $Q^i$, $i = 1, 2, \cdots, 5$, we first generate 10 noisy action-value samples independently (green dots in Figure 2(a)). Let $e^i(s, a)$ denote the *approximation error* of $Q^i$:

$$Q^i(s, a) = Q^*(s, a) + e^i(s, a), \ \text{ with } \ e^i(s, a) \sim \mathcal{U}(-\tau_i, \tau_i), \tag{4}$$

where $\tau_i \sim \mathcal{U}(0, \tau)$ models the approximation error distribution for the $i$-th approximator. Note that the assumption on the uniform error distribution is commonly used to indicate that both positive and negative approximation error are possible in Q-function approximators [29][17][6].

(II) Next, Figure 2(b) illustrates the ensemble ($N = 5$) of approximators for two actions, where each approximator is a 6-degree polynomial that fits the noisy values at sampled states.

(III) Following the same ensemble approach in [6][17], we randomly choose $M$ approximators from the ensemble and take the minimum over them to obtain a proxy approximator for each action, resulting in the dashed lines in Figure 2(c).

(IV) Finally, the maximum action value of the proxy approximator is used as the target to update the current approximators. To evaluate the target value *estimation error*, Figure 2(d) depicts the difference between the obtained target value and the underlying true value when using different ensemble size $M$. As in [33], we utilize the average estimation error (i.e., estimation bias) to quantify the performance of current approximators. For example, when the ensemble size $M = 2$, the red line is above zero for most states, implying the overestimation tendency in the target. Clearly, Figure 2(d) indicates that the estimation bias is highly dependent on the ensemble size, and even a change of $M$ can lead the shift from overestimation to underestimation. Since the Q-function approximation error of each approximator changes over time in the training process [5] (examples for this phenomenon

can be found in Appendix B.3), we next analyze the impact of the ensemble size on the estimation bias under different approximation error distributions. As shown in Figure 3(a), with a fixed ensemble size $M$, the estimation bias may shift between positive and negative and be 'dramatically' large for some error distributions. In light of this observation, departing from using a fixed size, we advocate to adapt the in-target ensemble size, e.g., set $M = 4$ when the noise parameter $\tau > 1.5$ and $M = 3$ otherwise. The estimation bias resulted by this adaptation mechanism is much closer to zero. Besides, Figure 3(b) characterizes the estimation bias under different action spaces, which is also important considering that different tasks normally have different action spaces and the number of available actions may vary in different states even for the same task. The adaptive ensemble approach is clearly more robust in our setting. In a nutshell, both Figure 3(a) and 3(b) suggest that a fixed ensemble size would not work well to minimize the estimation bias during learning for different tasks. This phenomenon has also been observed in the empirical results [17]. In stark contrast, adaptively changing the ensemble size based on the approximation error indeed can help to reduce the estimation bias in different settings.

## 3 Adaptive Ensemble Q-learning (AdaEQ)

Motivated by the illustrative example above, we next devise a generalized ensemble method with ensemble size adaptation to drive the estimation bias to be close to zero, by taking into consideration the time-varying feature of the approximation error during the learning process. Formally, we consider an ensemble of $N$ Q-function approximators, i.e., $\{Q_i\}_{i=1}^N$, with each approximator initialized independently and randomly. We use the minimum of a subset $\mathcal{M}$ of the $N$ approximators in the Q-learning target as in (3), where the size of subset $|\mathcal{M}| = M \leq N$.

### 3.1 Lower Bound and Upper Bound on Estimation Bias

We first answer the following key question:"*How does the approximation error, together with the ensemble size, impact the estimation bias*?". To this end, based on [29], we characterize the intrinsic relationship among the ensemble size $M$, the Q-function approximation error and the estimation bias, and derive an upper bound and a lower bound on the bias in the tabular case. Without loss of generality, we assume that for each state $s$, there are $A$ available actions.

Let $e^i(s,a) \triangleq Q^i(s,a) - Q^\pi(s,a)$ be the approximation error for the $i$-th Q-function approximator, where $Q^\pi(s,a)$ is the ground-truth of the Q-value for the current policy $\pi$. By using (3) to compute the target Q-value, we define the estimation error in the Bellman equation for transition $(s,a,r,s')$ as $Z_M$:
$$Z_M \triangleq r + \gamma \max_{a' \in \mathcal{A}} \min_{i \in \mathcal{M}} Q^i(s',a') - (r + \max_{a' \in \mathcal{A}} Q^\pi(s',a')).$$
Here a positive $\mathbf{E}[Z_M]$ implies overestimation bias while a negative $\mathbf{E}[Z_M]$ implies underestimation bias. Note that we use the subscription $M$ to emphasize that the estimation bias is intimately related to $M$.

**The case with two distributions for Q-function approximation errors.** For ease of exposition, we first consider the case when the approximation errors follow one of the two uniform distributions, as illustrated in Figure 4(a). Specifically, assume that for $i \in \mathcal{K} \subset \mathcal{M}$ with $|\mathcal{K}| = K$, $e^i(s,a) \sim \mathcal{U}(-\tau_1, \tau_1)$, and for $i \in \mathcal{M} \setminus \mathcal{K}$, $e^i(s,a) \sim \mathcal{U}(-\tau_2, \tau_2)$. Without loss of generality, we assume that $\tau_1 > \tau_2 > 0$. It is worth noting that in [29][17][6], the approximation error for all approximators is assumed to follow the same uniform distribution, i.e., $\tau_1 = \tau_2$, which is clearly more restrictive than the case here with two error distributions. For instance, when only one approximator is chosen to be updated at each step [17], the approximation error distribution of this approximator would change over time and hence differ from the others. We have the following results on the upper bound and lower bound of the estimation bias $\mathbf{E}[Z_M]$.

**Theorem 1.** *For the case with two distributions for Q-function approximation errors, the estimation bias $\mathbf{E}[Z_M]$ satisfies that*
$$\mathbf{E}[Z_M] \geq \gamma \left(\tau_1(1 - f_{AK} - 2f_{AM}) + \tau_2(1 - f_{AM})\right); \tag{5}$$
$$\mathbf{E}[Z_M] \leq \gamma \left(\tau_1 + \tau_2(1 - 2f_{A(M-K)} - (1 - \beta_K)^A)\right), \tag{6}$$
*where $\beta_K = (\frac{1}{2} - \frac{\tau_2}{2\tau_1})^K$, $f_{AK} = \frac{1}{K}B(\frac{1}{K}, A+1) = \frac{\Gamma(A+1)\Gamma(1+\frac{1}{K})}{\Gamma(A+\frac{1}{K}+1)} = \frac{A(A-1)\cdots 1}{(A+\frac{1}{K})(A+\frac{1}{K}-1)\cdots(1+\frac{1}{K})}$ with $B(\cdot,\cdot)$ being the Beta function.*

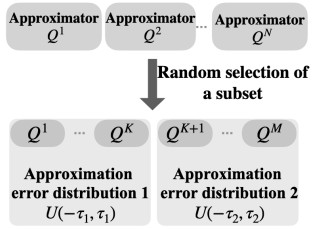

(a) Q-function approximation error distributions.

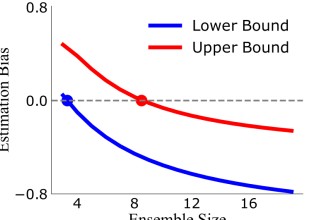

(b) Lower bound and upper bound on Estimation bias.

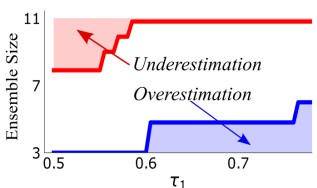

(c) Impact of approximation error on the estimation bias: overestimation vs. underestimation.

Figure 4: Illustration of upper bounds and lower bounds on estimation bias in Theorem 1. (a) The case where the approximation errors of the Q-approximators can be categorized into two uniform distributions. (b) The lower bound and the upper bound corresponding to (5) and (6), for given $\tau_1$, $\tau_2$, $A$: The blue point represents the 'critical' point where decreasing the ensemble size may lead overestimation (the lower bound is positive); and the red point denotes the 'critical' point where increasing ensemble size may lead underestimation (the upper bound is negative). (c) Due to time-varying feature of the approximation errors, the blue curve and the red curve depict the 'critical' points for the lower bound and the upper bound, respectively.

The proof of Theorem 1 is relegated to the Appendix A.1. Theorem 1 reveals that the estimation bias depends on the ensemble size as well as the approximation error distributions. To get a more concrete sense of Theorem 1, we consider an example where $\tau_1 = 0.5$ and $\tau_2 = 0.4$, as depicted in Figure 4(b), and characterize the relationship between the estimation bias and the ensemble size $M$. Notably, the estimation bias turns negative when the ensemble size $M > M_u = 9$ (red point: the value of $M$ where the upper bound is 0) and becomes positive when $M < M_l = 4$ (blue point: the value of $M$ where the lower bound is 0). In Figure 4(c), we fix $\tau_2 = 0.4$ and show how those two critical points ($M_u$ and $M_l$) change along with $\tau_1$. Here the red shaded area indicates underestimation bias when $M > M_u$, and the blue shaded area indicates overestimation bias when $M < M_l$. Clearly, in order to avoid the positive bias (blue shaded area), it is desirable to increase the ensemble size when the approximation error is large, e.g., $\tau_1 > 0.6$. On the other hand, decreasing the ensemble size is more preferred to avoid underestimation (red shaded area) when the approximation error is small, e.g., $\tau_1 < 0.6$.

**The general case with heterogeneous distributions for Q-function approximation errors.** Next, we consider a general case, in which the approximation errors for different approximators $\{Q^i\}$ are independently but non-identically distributed. Specifically, we assume that the approximation error $e^i(s, a)$ for $Q^i(s, a)$, $i = 1, 2, \cdots, M$, follows the uniform distribution $\mathcal{U}(-\tau_i, \tau_i)$, where $\tau_i > 0$. We use a multitude of tools to devise the upper bound and lower bound on the estimation bias $\mathbf{E}[Z_M]$. As expected, this general case is technically more challenging and the bounds would be not as sharp as in the special case with two distributions.

**Theorem 2.** *For the general case with heterogeneous error distributions, the estimation bias $\mathbf{E}[Z_M]$ satisfies that*

$$\mathbf{E}[Z_M] \geq \gamma \left( \tau_{\min} - \tau_{\max}(f_{A(M-1)} + 2f_{AM}) \right); \tag{7}$$

$$\mathbf{E}[Z_M] \leq \gamma \left( 2\tau_{\min} - \tau_{\max} \left( f_{AM} - 2g_{AM} \right) \right), \tag{8}$$

*where $\tau_{\min} = \min_i \tau_i$ and $\tau_{\max} = \max_i \tau_i$. $g_{AM} = \frac{1}{M} I_{0.5}(\frac{1}{M}, A + 1)$ with $I_{0.5}(\cdot, \cdot)$ being the regularized incomplete Beta function.*

Observe from Theorem 2 that the lower bound in (7) is positive when $\tau_{\min}(1 - 2f_{AM}) > \tau_{\max} f_{A(M-1)}$, indicating the existence of the overestimation issue. On thew contrary, the upper bound in (8) is negative when $2\tau_{\min} < \tau_{\max} (1 + f_{AM} - 2g_{AM})$, pointing to the underestimation issue. In general, when $\tau_{\min}$ is large enough, decreasing ensemble size $M$ is likely to cause overestimation, e.g., $\mathbf{E}[Z_M] \geq 0$ when $M < 2$. On the other hand, when $\tau_{\max}$ is small enough, increasing ensemble size $M$ is likely to cause underestimation, e.g., $\mathbf{E}[Z_M] \leq 0$ when $M$ is sufficiently large.

**Determination of parameter $c$.** As illustrated in Figure 4(c), for given approximation error characterization, a threshold $c$ can be chosen such that increasing the ensemble size would help to correct the overestimation bias when $\tau_{\max} > c$, and decreasing the ensemble size is more conductive to mitigate

the underestimation bias when $\tau_{\max} < c$. Specifically, parameter $c$ is determined in two steps. *Step 1: To estimate approximation error distribution parameters $\tau_{\min}$ and $\tau_{\max}$ by running an ensemble based algorithm (e.g., Algorithm 1) for a few epochs with a fixed ensemble size.* In particular, a testing trajectory is generated from a random initial state using the current policy to compute the (discounted) MC return $Q^\pi$ and the estimated Q-function value $Q^i, i = 1, 2, \cdots, N$. We next fit a uniform distribution model $\mathcal{U}(-\tau_i, \tau_i)$ of the approximation error $(Q^i - Q^\pi)$ for each Q-function approximator $Q^i$. Then, $\tau_{\min}$ and $\tau_{\max}$ can be obtained by choosing the minimum and maximum values among $\tau_i, i = 1, 2, \cdots, N$. *Step 2: To obtain the upper bound and the lower bound in Theorem 2 by using $\{\tau_{\min}, \tau_{\max}, A, \gamma\}$.* We investigate the relationship between ensemble size $M$ and the estimation bias by studying the bounds and identifying the 'critical' points as illustrated in Figure 4(b). Observe that a 'proper' ensemble size should be chosen between the 'critical' points, so as to reduce the overestimation and underestimation bias as much as possible. Since the approximation error is time-varying during the learning process, these two 'critical' points vary along with $\{\tau_{\max}\}$ and $\{\tau_{\min}\}$ (as shown in Figure 4(c)). Intuitively, it is desirable to drive the system to avoid both the red region (underestimation) and the blue region (overestimation). It can be clearly observed that there is a wide range of choice for parameter $c$ (e.g., $[0.5, 0.7]$ in Figure 4(c)) for the algorithm to stay in the white region, indicating that even though the pre-determined $c$ above is not optimized, it can still serve the purpose well.

The proof of Theorem 2 and numerical illustration can be found in the Appendix A.3. Summarizing, both Theorem 1 and Theorem 2 indicate that the approximation error characterization plays a critical role in controlling the estimation bias. In fact, both the lower bound and the upper bound in Theorem 2 depends on $\tau_{\min}$ and $\tau_{\max}$, which are time-varying due to the iterative nature of the learning process, indicating that it is sensible to use an adaptive ensemble size to drive the estimation bias to be close to zero, as much as possible.

## 3.2   Practical Implementation

Based on the theoretic findings above, we next propose AdaEQ that adapts the ensemble size based on the approximation error feedback on the fly, so as to drive the estimation bias close to zero. Particularly, as summarized in Algorithm 1, AdaEQ introduces two important steps at each iteration $t$, i.e., approximation error characterization (line 3) and ensemble size adaptation (line 4), which can be combined with the framework of either Q-learning or actor-critic methods.

**Characterization of the time-varying approximation error.** As outlined in Algorithm 1, the first key step is to quantify the time-varying approximation error at each iteration $t$ (for ease of exposition, we omit the subscript $t$ when it is clear from the context). Along same line as in [9, 33, 6], we run a testing trajectory of length $H$, $\mathcal{T} = (s_0, a_0, s_1, a_1, \cdots, s_H, a_H)$, from a random initial state using the current policy $\pi$, and compute the discounted Monte Carlo return $Q^\pi(s, a)$ and the estimated Q-function value $Q^i(s, a)$, $i = 1, \cdots, N$ for each visited state-action pair $(s, a)$. The empirical standard derivation of $Q^i(s, a) - Q^\pi(s, a)$ can be then obtained to quantify the approximation error of each approximator $Q^i$. Then, we take the average of the empirical standard derivation over all approximators to characterize the approximation error at the current iteration $t$, i.e.,

$$\tilde{\tau}_t = \frac{1}{N} \sum_{i=1}^N \mathrm{std}(Q^i(s, a) - Q^\pi(s, a)), \ \ (s, a) \in \mathcal{T}. \tag{9}$$

**Error-feedback based ensemble size adaptation.** Based on the theoretic results and Figure 4(c), we update the ensemble size $M$ at each iteration $t$ based on the approximation error (9), using the following piecewise function:

$$M_t = \begin{cases} \mathrm{rand}(M_{t-1}+1, N) & \tilde{\tau}_{t-1} > c, \ M_{t-1}+1 \le N \\ \mathrm{rand}(2, M_{t-1}-1) & \tilde{\tau}_{t-1} < c, \ M_{t-1}-1 \ge 2 \\ M_{t-1} & \text{otherwise}, \end{cases} \tag{10}$$

where $\mathrm{rand}(\cdot, \cdot)$ is a uniform random function and $c$ is a pre-determined parameter to capture the 'tolerance' of the estimation bias during the adaptation process. Recall that parameter $c$ can be determined by using the upper bound and the lower bound in Theorem 2 (Theorem 1). Particularly, a larger $c$ implies that more tolerance of the underestimation bias is allowed when adapting the ensemble size $M_t$. A smaller $c$, on the other hand, admits more tolerance of the overestimation. In this way, AdaEQ can be viewed as a generalization of Maxmin and REDQ with ensemble size adaptation. In particular, when $c = 0$ and $M_t + 1 \le N$, the adaptation mechanism would increase the ensemble size until it is equal to $N$. Consequently, AdaEQ degenerates to Maxmin [17] where $M = N$, leading

**Algorithm 1** Adaptive Ensemble Q-learning (AdaEQ)

---

1: Empty replay buffer $\mathcal{D}$, step size $\alpha$, number of the approximators $N$, initial in-target ensemble size $M_0 \leq N$, initial state $s$. Initialize $N$ approximators with different training samples.
2: **for** Iteration $t = 1, 2, 3, \cdots$ **do**
3:     Identify approximation error parameter $\tilde{\tau}_t$ using (9)
4:     Update ensemble size $M_t$ according to (10)
5:     Sample a set $\mathcal{M}$ of $M_t$ different indices from $\{1, 2, \cdots, N\}$
6:     Obtain the proxy approximator $Q^{\text{proxy}}(s, a) \leftarrow \min_{i \in \mathcal{M}} Q^i(s, a), \forall a \in \mathcal{A}$
7:     Choose action $a$ from current state $s$ using policy derived from $Q^{\text{proxy}}$ (e.g., $\varepsilon$-greedy)
8:     Take action $a$, observe $r$ and next state $s'$
9:     Update replay buffer $\mathcal{D} \leftarrow \mathcal{D} \cup \{s, a, r, s'\}$
10:     **for** $i = 1, 2, \cdots, N$ **do**
11:         Sample a random mini-batch $B$ from $\mathcal{D}$
12:         Compute the target: $y(s, a, r, s') \leftarrow r + \gamma \max_{a' \in \mathcal{A}} Q^{\text{proxy}}(s', a'), (s, a, r, s') \in B$
13:         Update Q-function $Q^i$: $Q^i(s, a) \leftarrow (1 - \alpha)Q^i(s, a) + \alpha y(s, a, r, s'), (s, a, r, s') \in B$
14:     **end for**
15:     $s \leftarrow s'$
16: **end for**

---

to possible underestimation bias. Meantime, when $c$ is set sufficiently large, the ensemble size $M$ would decrease until reaching the minimal value 2 during the learning process, where the estimation bias would be positive according to Theorem 2. In this case, AdaEQ is degenerated to REDQ [6] with ensemble size $M = 2$. We show the convergence analysis of AdaEQ in Appendix A.5.

**Remark.** We use random sampling in Eqn. (10) for two reasons. Firstly, the characterization of the approximation error in Eqn. (9) is noisy in nature. In particular, Monte Carlo returns with finite-length testing trajectory may introduce empirical errors when estimating the underlying ground true value of $Q^\pi$. This noisy estimation is often the case when the policy is not deterministic, or the environment is not deterministic. Thus, we use random sampling to 'capture' the impact of this noisy estimation. Secondly, in general it is infeasible to characterize the exact relationship between estimation bias $Z_M$ and ensemble size $M$. Without any further prior information except from the bounds we obtained in Theorem 1 and Theorem 2 about the approximation error, the random sampling can be viewed as the 'exploration' in AdaEQ.

## 4 Experimental Results

In this section, we evaluate the effectiveness of AdaEQ by answering the following questions: 1) Can AdaEQ minimize the estimation bias and further improve the performance in comparison to existing ensemble methods? 2) How does AdaEQ perform given different initial ensemble sizes? 3) How does the 'tolerance' parameter $c$ affect the performance?

To make a fair comparison, we follow the setup of [6] and use the same code base to compare the performance of AdaEQ with REDQ [6] and Average-DQN (AVG) [2], on three MuJoCo continuous control tasks: Hopper, Ant and Walker2d. The same hyperparameters are used for all the algorithms. Specifically, we consider $N = 10$ Q-function approximators in total. The ensemble size $M = N = 10$ for AVG, while the initial $M$ for AdaEQ is set as 4. The ensemble size for REDQ is set as $M = 2$, which is the fine-tuned result from [6]. For all the experiments, we set the 'tolerance' parameter $c$ in (10) as 0.3 and the length of the testing trajectories as $H = 500$. The ensemble size is updated according to (10) every 10 epochs in AdaEQ. The discount factor is 0.99. Implementation details and hyperparamter settings are fully described in Appendix B.1.

**Evaluation of estimation bias.** To investigate the impact of the adaptation mechanism in AdaEQ, we begin by examining how the estimation bias changes in the training process. After each epoch, we run an evaluation episode of length $H = 500$, starting from an initial state sampled from the replay buffer. We calculate the estimation error based on the difference between the Monte Carlo return value and the Q-estimates as in [33, 6, 14]. For each experiment, the shaded area represents a standard deviation of the average evaluation over 3 training seeds. As shown in the first row of Figure 5, AdaEQ can reduce the estimation bias to nearly zero in all three benchmark environments, in contrast to REDQ and AVG. The AVG approach tends to result in positive bias in all three environments

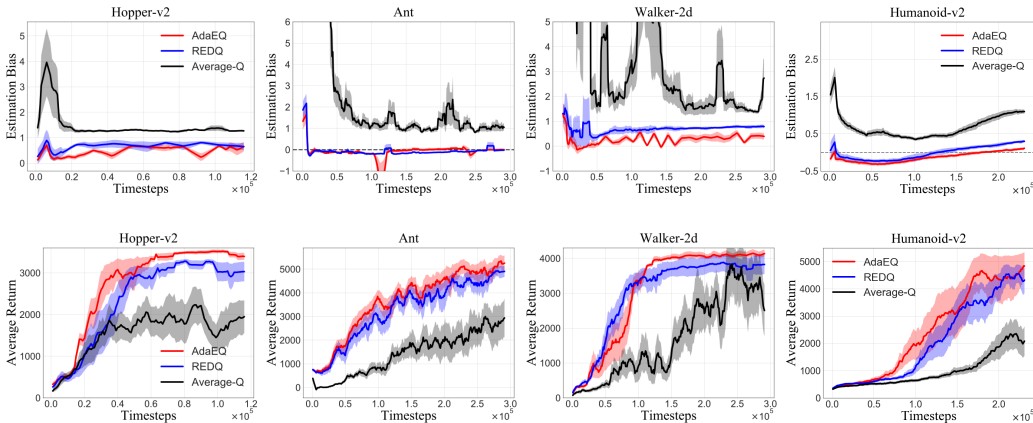

Figure 5: Comparison in terms of the estimation bias (first row) and average return (second row) in three MuJoCo tasks. Solid lines are the mean values and the shaded areas are the standard derivations across three random seeds. We use the undiscounted sum of all the reward in the testing episode to evaluate the performance of the current policy after each epoch. The estimation error is evaluated by comparing the difference between the Monte Carlo return and the average Q-value for each state-action pair visited during the testing episode. We take the average of those error values as estimation bias. For AdaEQ, we use the same hyperparameter $c$ for ensemble size adaptation.

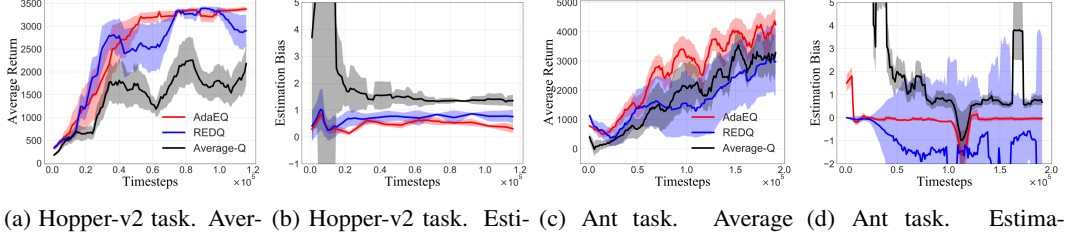

(a) Hopper-v2 task. Average returns over different initial ensemble size $M = 2, 3, 5$.

(b) Hopper-v2 task. Estimation bias over different initial ensemble size $M = 2, 3, 5$.

(c) Ant task. Average returns over different initial ensemble size $M = 3, 5, 7$.

(d) Ant task. Estimation bias over different initial ensemble size $M = 3, 5, 7$.

Figure 6: Impacts of the initial ensemble size $M$ on the performance of AdaEQ in Hopper-v2 and Ant task. The solid lines are the mean values and the shaded areas are the standard derivations across three ensemble size settings.

during the learning procedure, which is consistent with the results obtained in [6]. Notably, it can be clearly observed from Hopper and Walker2d tasks that the estimation bias for AdaEQ is driven to be close to zero, thanks to the dynamic ensemble size adjustment during the learning process. Meantime, in the Ant task, even though the fine-tuned REDQ can mitigate the overestimation bias, it tends to have underestimation bias, whereas AdaEQ is able to keep the bias closer to zero (gray dashed line) even under a 'non-optimal' choice of the initial ensemble size.

**Performance on MuJoCo benchmark.** We evaluate the policy return after each epoch by calculating the undiscounted sum of rewards when running the current learnt policy [6, 14]. The second row of Figure 5 demonstrates the average return during the learning process for AdaEQ, AVG and REDQ, respectively. Especially, we choose the fine-tune ensemble size for REDQ [6]. As observed in Figure 5, AdaEQ can efficiently learn a better policy and achieve higher average return in all three challenging MuJoCo tasks, without searching the optimal parameters beforehand for each of them. Meantime, AdaEQ only incurs slightly more computation time than REDQ in most MuJoCo tasks. Due to space limitation, we have relegated the wall-clock training time comparison to Table 2 in Appendix B.2.

**Robustness to the initial ensemble size.** Next, we investigate the performance of AdaEQ under different settings of the initial ensemble size in the Hopper-v2 and Ant environment, i.e., $M = (2, 3, 5)$

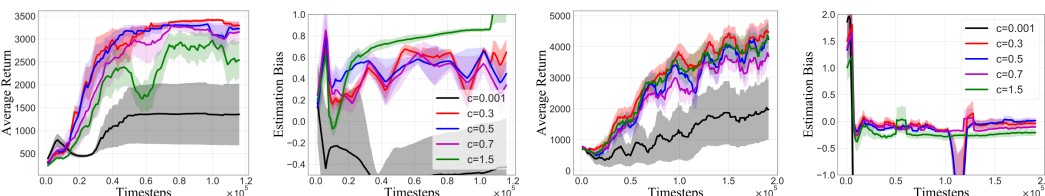

(a) Hopper-v2 task. Average returns over different parameter $c$ in AdaEQ. (b) Hopper-v2 task. Estimation bias over different parameter $c$ in AdaEQ. (c) Ant task. Average returns over different parameter $c$ in AdaEQ. (d) Ant task. Estimation bias over different parameter $c$ in AdaEQ.

Figure 7: Impacts of parameter $c$ on the performance of AdaEQ in Hopper-v2 and Ant task. The initial ensemble size is set to be $M = 4$. The mean value and the standard derivation are evaluated across three training seeds.

and $M = (3, 5, 7)$. As shown in Figure 6, AdaEQ consistently outperforms the others in terms of the average performance over different setups, which implies the benefit of adjusting the in-target ensemble size based on the error feedback. It can be seen from the shaded area that the performance of AVG and REDQ, may vary significantly when the ensemble size changes.

**Robustness to parameter $c$ in a wide range.** As illustrated in Figure 7, we conduct the ablation study by setting $c = 0.001, 0.3, 0.5, 0.7, 1.5$ on the Hopper-v2 and Ant tasks. Clearly, AdaEQ works better for $c \in [0.3, 0.7]$. The experiment results corroborate our analysis in Section 3.1 that our algorithm is not sensitive to parameter $c$ in a wide range. As mentioned in Section 3.2, when parameter $c$ is close to zero, AdaEQ degenerates to Maxmin, which is known to suffer from underestimation bias when the ensemble size is large [6]. Further, as illustrated in Figure 7(b), when $c$ is large, e.g., $c = 1.5$, the ensemble size would gradually decrease to the minimum and hence would not be able to throttle the overestimation tendency during the learning process.

## 5    Conclusion

Determining the right ensemble size is highly nontrivial for the ensemble Q-learning to correct the overestimation without introducing significant underestimation bias. In this paper, we devise AdaEQ, a generalized ensemble Q-learning method for the ensemble size adaptation, aiming to minimize the estimation bias during the learning process. More specifically, by establishing the upper bound and the lower bound of the estimation bias, we first characterize the impact of both the ensemble size and the time-varying approximation error on the estimation bias. Building upon the theoretic results, we treat the estimation bias minimization as an adaptive control problem, and take the approximation error as feedback to adjust the ensemble size adaptively during the learning process. Our experiments show that AdaEQ consistently and effectively outperforms the existing ensemble methods, such as REDQ and AVG in MuJoCo tasks, corroborating the benefit of using AdaEQ to drive the estimation bias close to zero.

There are many important avenues for future work. In terms of the bounds of the estimation bias, our analysis builds upon the standard independent assumption as in previous works. It's worth noting that in practice, the errors are generally correlated [29] and the theoretical analysis for this case remains not well understood. Additionally, in this work, we use a heuristic tolerance parameter in the adaptation mechanism to strike the balance in controlling the positive bias and negative bias. It is of great interest to develop a systematic approach to optimize this tolerance parameter.

## Acknowledgments and Disclosure of Funding

We thank the anonymous reviewers for their constructive comments. This work was supported in part by NSF grants CNS-2130125, CCSS-2121222 and CNS-2003081.

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
