# Appendix

## A  Proofs

### A.1  The Proof of Theorem 1

We first restate the following results from order statistics [17].

**Lemma 1.** *Let $X_1, X_2, \cdots, X_M$ be $M$ i.i.d random variables from an absolutely continuous distribution with probability density function (PDF) $f(x)$ and cumulative distribution function (CDF) $F(x)$. Denote $\mu = \mathbf{E}[X_i]$ and $\sigma^2 = Var[X_i] < +\infty$. Let $X_{1:M} \leq X_{2:M} \leq \cdots \leq X_{M:M}$ be the order statistics obtained by reordering these random variables in increasing order of magnitude. Denote the PDF and CDF of $X_{1:M} = \min_{i \in \mathcal{M}} X_i$ as $f_{1:M}(x)$ and $F_{1:M}(x)$. Denote $f_{M:M}(x)$ and $F_{M:M}(x)$ to be the PDF and CDF of $X_{M:M} = \max_{i \in \mathcal{M}} X_i$. Then we have*

*(i)  $f_{1:M}(x) = Mf(x)(1 - F(x))^{M-1}$,  $F_{1:M}(x) = 1 - (1 - F(x))^M$.*

*(ii)  $f_{M:M}(x) = Mf(x)(F(x))^{M-1}$,  $F_{M:M}(x) = (F(x))^M$.*

Now we prove Theorem 1.

**Theorem 1.** *For the case with two distributions for Q-function approximation errors, the estimation bias $\mathbf{E}[Z_M]$ satisfies that*

$$\mathbf{E}[Z_M] \geq \gamma \left( \tau_1(1 - f_{AK} - 2f_{AM}) + \tau_2(1 - f_{AM}) \right); \tag{5}$$

$$\mathbf{E}[Z_M] \leq \gamma \left( \tau_1 + \tau_2(1 - 2f_{A(M-K)} - (1 - \beta_K)^A) \right), \tag{6}$$

*where $\beta_K = (\frac{1}{2} - \frac{\tau_2}{2\tau_1})^K$, $f_{AK} = \frac{1}{K}B(\frac{1}{K}, A+1) = \frac{\Gamma(A+1)\Gamma(1+\frac{1}{K})}{\Gamma(A+\frac{1}{K}+1)} = \frac{A(A-1)\cdots 1}{(A+\frac{1}{K})(A+\frac{1}{K}-1)\cdots(1+\frac{1}{K})}$ with $B(\cdot, \cdot)$ being the Beta function.*

*Proof.* Assume that for $i \in \mathcal{K} \subset \mathcal{M}$, $e^i(s, a) \sim \mathcal{U}(-\tau_1, \tau_1)$ with PDF $f_1(x)$ and CDF $F_2(x)$, and for $i \in \mathcal{M} \setminus \mathcal{K}$, $e^i(s, a) \sim \mathcal{U}(-\tau_2, \tau_2)$ with PDF $f_2(x)$ and CDF $F_2(x)$. Without loss of generality, we assume that $\tau_1 > \tau_2 > 0$.

From Lemma 1, it is clear that

$$F_{1:M}(x) = \mathbf{P}(X_{1:M} \leq x) = \begin{cases} 1 - (1 - F_1(x))^K & x < -\tau_2 \\ 1 - (1 - F_1(x))^K(1 - F_2(x))^{M-K} & x \in [-\tau_2, \tau_2] \\ 1 & x > \tau_2 \end{cases}, \tag{11}$$

$$f_{1:M}(x) = \frac{dF_{1:M}(x)}{dx},$$

where $F_1(x) = \frac{1}{2} + \frac{x}{2\tau_1}$, $f_1(x) = \frac{1}{2\tau_1}$, $x \in [-\tau_1, \tau_1]$ and $F_2(x) = \frac{1}{2} + \frac{x}{2\tau_2}$, $f_2(x) = \frac{1}{2\tau_2}$, $x \in [-\tau_2, \tau_2]$. Denote $F_{1:K}(x) = 1 - (1 - F_1(x))^K$ and $f_{1:K}(x) = Kf_1(x)(1 - F_1(x))^{K-1}$. Then, the estimation bias can be obtained as

$$\mathbf{E}[Z_M] = \gamma \, \mathbf{E}[\max_{a'} \min_{i \in \mathcal{M}} e^i(s, a)]$$

$$= \gamma \int_{-\tau_1}^{\tau_1} Ax f_{1:M}(x) F_{1:M}(x)^{A-1} dx$$

$$= \gamma (\underbrace{\int_{-\tau_1}^{-\tau_2}}_{①} + \underbrace{\int_{-\tau_2}^{\tau_2}}_{②} + \underbrace{\int_{\tau_2}^{\tau_1}}_{③}) dx. \tag{12}$$

We first consider term ① in (12), where

$$① = \int_{-\tau_1}^{-\tau_2} Ax f_{1:K}(x) F_{1:K}(x)^{A-1} dx$$

$$= (\int_{-\tau_1}^{\tau_1} - \int_{-\tau_2}^{\tau_2} - \int_{\tau_2}^{\tau_1}) Ax f_{1:K}(x) F_{1:K}(x)^{A-1} dx.$$

It follows that ① can be bounded above as follows:

$$① \leq \left( \int_{-\tau_1}^{\tau_1} - \int_{-\tau_2}^{\tau_2} \right) Ax f_{1:K}(x) F_{1:K}(x)^{A-1} dx$$

$$= \int_{-\tau_1}^{\tau_1} AxK f_1(x)(1 - F_1(x))^{K-1} \left( 1 - (1 - F_1(x))^K \right)^{A-1} dx$$

$$- \int_{-\tau_2}^{\tau_2} AxK f_1(x)(1 - F_1(x))^{K-1} \left( 1 - (1 - F_1(x))^K \right)^{A-1} dx$$

$$= \tau_1 [1 - 2f_{AK}] - \tau_2 \left( 1 - (\frac{1}{2} - \frac{\tau_2}{2\tau_1})^K \right)^A - \tau_2 \left( 1 - (\frac{1}{2} + \frac{\tau_2}{2\tau_1})^K \right)^A + \underbrace{2\tau_1 \int_{\frac{1}{2} - \frac{\tau_2}{2\tau_1}}^{\frac{1}{2} + \frac{\tau_2}{2\tau_1}} (1 - y^K)^A dy}_{\leq f_{AK}}$$

$$\leq \tau_1 - \tau_2 \left( 1 - (\frac{1}{2} - \frac{\tau_2}{2\tau_1})^K \right)^A - \tau_2 \left( 1 - (\frac{1}{2} + \frac{\tau_2}{2\tau_1})^K \right)^A,$$

where $f_{AK} = \frac{1}{K} B(\frac{1}{K}, A+1) = \frac{\Gamma(A+1)\Gamma(1+\frac{1}{K})}{\Gamma(A+\frac{1}{K}+1)} = \frac{A(A-1)\cdots 1}{(A+\frac{1}{K})(A+\frac{1}{K}-1)\cdots(1+\frac{1}{K})}$ with $B(\cdot, \cdot)$ being beta function and $y := \frac{1}{2} - \frac{x}{2\tau_1}$.

We can also have the following lower bound on ①:

$$① = \tau_1 (1 - 2f_{AK}) - \tau_2 \left( 1 - (\frac{1}{2} - \frac{\tau_2}{2\tau_1})^K \right)^A - \tau_2 \left( 1 - (\frac{1}{2} + \frac{\tau_2}{2\tau_1})^K \right)^A + 2\tau_1 \int_{\frac{1}{2} - \frac{\tau_2}{2\tau_1}}^{\frac{1}{2} + \frac{\tau_2}{2\tau_1}} (1 - y^K)^A dy$$

$$+ \tau_2 \left( 1 - (\frac{1}{2} - \frac{\tau_2}{2\tau_1})^K \right)^A + 2\tau_1 \int_0^{\frac{1}{2} - \frac{\tau_2}{2\tau_1}} (1 - y^K)^A dy$$

$$= \tau_1 (1 - 2f_{AK}) - \tau_2 \left( 1 - (\frac{1}{2} + \frac{\tau_2}{2\tau_1})^K \right)^A + 2\tau_1 \int_0^{\frac{1}{2} + \frac{\tau_2}{2\tau_1}} (1 - y^K)^A dy$$

$$\geq \tau_1 (1 - 2f_{AK}) - \tau_2 \left( 1 - (\frac{1}{2} + \frac{\tau_2}{2\tau_1})^K \right)^A + \tau_1 f_{AK}$$

$$= \tau_1 (1 - f_{AK}) - \tau_2 \left( 1 - (\frac{1}{2} + \frac{\tau_2}{2\tau_1})^K \right)^A.$$

For the term ② in (12), it follows that

$$② = \int_{-\tau_2}^{\tau_2} Ax f_{1:M}(x) F_{1:M}(x)^{A-1} dx$$

$$= \int_{-\tau_2}^{\tau_2} xd \left[ 1 - (\frac{1}{2} - \frac{x}{2\tau_1})^K (\frac{1}{2} - \frac{x}{2\tau_2})^{M-K} \right]^A dx$$

$$= \tau_2 + \tau_2 \left( 1 - (\frac{1}{2} + \frac{\tau_2}{2\tau_1})^K \right)^A - \underbrace{\int_{-\tau_2}^{\tau_2} \left[ 1 - (\frac{1}{2} - \frac{x}{2\tau_1})^K (\frac{1}{2} - \frac{x}{2\tau_2})^{M-K} \right]^A dx}_{(*)},$$

where $(*)$ satisfies that

$$(*) > \int_{-\tau_2}^{\tau_2} \left[ 1 - (\frac{1}{2} - \frac{x}{2\tau_2})^{M-K} \right]^A dx = 2\tau_2 f_{A(M-K)},$$

$$(*) = \int_{-\tau_2}^0 + \int_0^{\tau_2}$$

$$< \int_{-\tau_2}^0 [1 - (\frac{1}{2} - \frac{x}{2\tau_2})^M]^A dx + \int_0^{\tau_2} [1 - (\frac{1}{2} - \frac{x}{2\tau_1})^M]^A dx$$

$$< \tau_2 f_{AM} + 2\tau_1 f_{AM}.$$

From the definition of $F_{1:M}(x)$, we have that term ③ in (12) equals to zero.

Summarizing, we obtain the following upper bound and lower bound on the estimation bias $\mathbf{E}[Z_M]$, for the case with two distributions for Q-function approximation errors:

$$\mathbf{E}[Z_M] \geq \gamma \left( \tau_1(1 - f_{AK} - 2f_{AM}) + \tau_2(1 - 2f_{AM}) \right)$$

$$\mathbf{E}[Z_M] \leq \gamma \left( \tau_1 + \tau_2[1 - 2f_{A(M-K)}] - \tau_2(1 - \beta_K)^A \right)$$

where $\beta_K = (\frac{1}{2} - \frac{\tau_2}{2\tau_1})^K$. □

## A.2 Parameter Setting in Numerical Illustration of Figure 4

In Figure 4(a), the approximation error parameter $\tau_1 = 0.5$, $\tau_2 = 0.4$ and the number of actions is set as $A = 30$. The number of approximators for which the approximation errors follow the first distribution is $K = 2$. In Figure 4(c), we fix $\tau_2 = 0.4$, $A = 30$, $K = 2$ and $\tau_1$ ranges from 0.5 to 0.8. The changes of the 'critical' points (red dot and blue dot) are depicted in Figure 4(c).

## A.3 The Proof of Theorem 2

**Theorem 2.** *For the general case with heterogeneous error distributions, the estimation bias $\mathbf{E}[Z_M]$ satisfies that*

$$\mathbf{E}[Z_M] \geq \gamma \left( \tau_{\min} - \tau_{\max}(f_{A(M-1)} + 2f_{AM}) \right); \tag{7}$$

$$\mathbf{E}[Z_M] \leq \gamma \left( 2\tau_{\min} - \tau_{\max}(f_{AM} - 2g_{AM}) \right), \tag{8}$$

*where $\tau_{\min} = \min_i \tau_i$ and $\tau_{\max} = \max_i \tau_i$. $g_{AM} = \frac{1}{M} I_{0.5}(\frac{1}{M}, A + 1)$ with $I_{0.5}(\cdot, \cdot)$ being the regularized incomplete Beta function.*

*Proof.* Assume that the approximation error $e^i(s, a)$ for each Q approximator follows uniform distribution $\mathcal{U}(-\tau_i, \tau_i)$ with PDF $f_i(x)$ and CDF $F_i(x)$. The PDF and CDF for each approximation error distribution is as follows,

$$f_i(x) = \begin{cases} \frac{1}{2\tau_i}, & x \in [-\tau_i, \tau_i] \\ 0, & \text{otherwise} \end{cases}, \quad F_i(x) = \begin{cases} 0, & x < -\tau_i \\ \frac{x+\tau_i}{2\tau_i}, & x \in [-\tau_i, \tau_i] \\ 1, & x > \tau_i \end{cases}$$

From Lemma 1, it can be seen that

$$F_{1:M}(x) = \mathbf{P}(X_{1:M} \leq x) = 1 - \prod_{i=1}^{M}(1 - F_i(x)),$$

$$f_{1:M}(x) = \sum_{i=1}^{M} \left( f_i(x) \prod_{j \neq i}(1 - F_j(x)) \right).$$

Assume that at state $s'$, there are $A$ actions applicable. Then the estimation bias $Z_M$ is

$$\mathbf{E}[Z_M] = \gamma \mathbf{E}_{\tau_1, \cdots, \tau_M} \left[ \max_{a'} \min_{i \in \mathcal{M}} e^i(s, a) \right]$$

$$= \gamma \left[ \int_{-\tau_{\max}}^{\tau_{\max}} Ax f_{1:M}(s) F_{1:M}(x)^{A-1} dx \right].$$

Considering the integration terms, we conclude that

$$\int_{-\tau_{\max}}^{\tau_{\max}} Ax f_{1:M}(s) F_{1:M}(x)^{A-1} dx$$

$$= \int_{-\tau_{\max}}^{\tau_{\max}} x d(F_{1:M}(x)^A)$$

$$= x(F_{1:M}(x)^A)\big|_{-\tau_{\max}}^{\tau_{\max}} - \int_{-\tau_{\max}}^{\tau_{\max}} F_{1:M}(x)^A dx$$

$$= \tau_{\max} - \int_{-\tau_{\max}}^{\tau_{\max}} \left( 1 - \prod_{i=1}^{M}(1 - F_i(x)) \right)^A dx$$

$$= \tau_{\max} - \underbrace{\int_{-\tau_{\max}}^{\tau_{\max}} (1 - (1 - F_1(x))(1 - F_2(x)) \cdots (1 - F_M(x)))^A \, dx,}_{\textcircled{1}}$$

where $\tau_{\min} = \min_i \tau_i$ and $\tau_{\max} = \max_i \tau_i$. Denote $f_{AM} = \frac{1}{M} B(\frac{1}{M}, A+1) = \frac{\Gamma(A+1)\Gamma(1+\frac{1}{M})}{\Gamma(A+\frac{1}{M}+1)} = \frac{A(A-1)\cdots 1}{(A+\frac{1}{M})(A+\frac{1}{M}-1)\cdots(1+\frac{1}{M})}$ with $B(\cdot, \cdot)$ being beta function.

It first can be seen that $\textcircled{1}$ satisfies that

$$\textcircled{1} \leq \int_{-\tau_{\min}}^{0} \left(1 - (\frac{1}{2} - \frac{1}{2\tau_{\max}}x)^M\right)^A dx + \int_{0}^{\tau_{\min}} \left(1 - (\frac{1}{2} - \frac{1}{2\tau_{\min}}x)^M\right)^A dx$$

$$+ \int_{\tau_{\min}}^{\tau_{\max}} dx + \int_{-\tau_{\max}}^{-\tau_{\min}} (1 - (\frac{1}{2} - \frac{x}{2\tau_{\max}})^{M-1})^A dx$$

$$= 2\tau_{\max} \int_{\frac{1}{2}}^{\frac{1}{2}+\frac{\tau_{\min}}{2\tau_{\max}}} (1 - y^M)^A dy + 2\tau_{\min} \int_{0}^{\frac{1}{2}} (1 - y^M)^A dy$$

$$+ \tau_{\max} - \tau_{\min} + 2\tau_{\max} \int_{\frac{1}{2}+\frac{\tau_{\min}}{2\tau_{\max}}}^{1} (1 - y^{M-1})^A dy \quad (y := \frac{1}{2} - \frac{x}{2\tau_{\max}})$$

$$\leq 2\tau_{\max} \int_{0}^{1} (1 - y^M)^A dy + \tau_{\max} - \tau_{\min} + \tau_{\max} f_{A(M-1)}$$

$$\leq 2\tau_{\max} f_{AM} + \tau_{\max} - \tau_{\min} + \tau_{\max} f_{A(M-1)}.$$

And the lower bound on $\textcircled{1}$ can be obtained as

$$\textcircled{1} \geq \int_{-\tau_{\min}}^{0} \left(1 - (\frac{1}{2} - \frac{1}{2\tau_{\min}}x)^M\right)^A dx + \int_{0}^{\tau_{\min}} \left(1 - (\frac{1}{2} - \frac{1}{2\tau_{\max}}x)^M\right)^A dx$$

$$+ \tau_{\max} - \tau_{\min} + \int_{-\tau_{\max}}^{-\tau_{\min}} (1 - (\frac{1}{2} - \frac{x}{2\tau_{\max}}))^A dx$$

$$= 2\tau_{\min} \int_{\frac{1}{2}}^{1} (1 - y^M)^A dy + 2\tau_{\max} \int_{\frac{1}{2}-\frac{1}{2}\frac{\tau_{\min}}{\tau_{\max}}}^{\frac{1}{2}} (1 - y^M)^A dy + \tau_{\max} - \tau_{\min} + \frac{\tau_{\max}(1 - \tau_{\min}/\tau_{\max})^{A+1}}{2^A(A+1)}$$

$$\geq \tau_{\min} \frac{2}{M} I_{0.5}(\frac{1}{M}, A+1) + 2\tau_{\max} \left(\int_{0}^{1} - \int_{\frac{1}{2}}^{1}\right) (1 - y^M)^A dy + \tau_{\max} - \tau_{\min} + \frac{\tau_{\max}(1 - \tau_{\min}/\tau_{\max})^{A+1}}{2^A(A+1)}$$

$$\geq 2\tau_{\min} g_{AM} + 2\tau_{\max} f_{AM} - 2\tau_{\max} g_{AM} + \tau_{\max} - \tau_{\min} + \frac{\tau_{\max}(1 - \tau_{\min}/\tau_{\max})^{A+1}}{2^A(A+1)},$$

where $g_{AM} = \frac{1}{M} I_{0.5}(\frac{1}{M}, A+1)$ with $I_{0.5}(\cdot, \cdot)$ being the regularized incomplete Beta function. The last inequality is true due to the following result:

$$\int_{\frac{1}{2}}^{1} (1 - y^M)^A dy = \frac{1}{M} \int_{(\frac{1}{2})^M}^{1} t^{\frac{1}{M}-1}(1 - t)^A dt$$

$$\geq \frac{1}{M} \int_{\frac{1}{2}}^{1} t^{\frac{1}{M}-1}(1 - t)^A dt$$

$$= \frac{1}{M} I_{0.5}(\frac{1}{M}, A+1) \quad (t := y^M)$$

$$:= g_{AM}.$$

Consequently, we have that $\textcircled{1}$ satisfies

$$\textcircled{1} \geq \tau_{\max} - \tau_{\min} + \frac{\tau_{\max}(1 - \tau_{\min}/\tau_{\max})^{A+1}}{2^A(A+1)} + 2\tau_{\min} g_{AM} + 2\tau_{\max} f_{AM} - 2\tau_{\max} g_{AM}$$

$$\geq \tau_{\max} - \tau_{\min} + \frac{\tau_{\max} - (A+1)\tau_{\min}}{2^A(A+1)} + \tau_{\max} f_{AM} - 2\tau_{\max} g_{AM} + 2\tau_{\min} g_{AM}$$

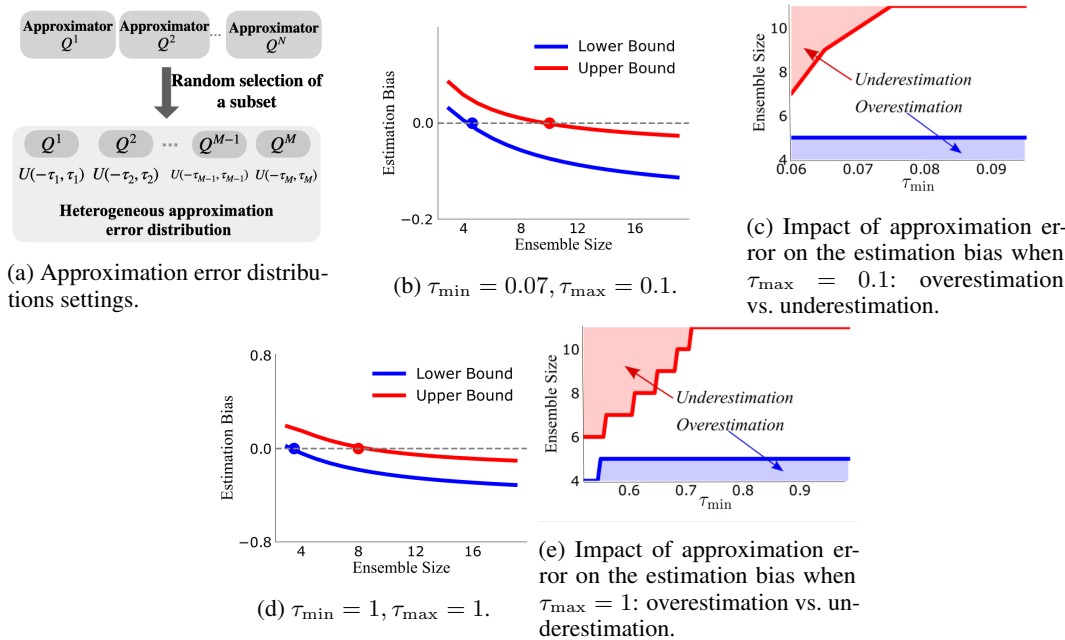

(a) Approximation error distributions settings.

(b) $\tau_{\min} = 0.07, \tau_{\max} = 0.1$.

(c) Impact of approximation error on the estimation bias when $\tau_{\max} = 0.1$: overestimation vs. underestimation.

(d) $\tau_{\min} = 1, \tau_{\max} = 1$.

(e) Impact of approximation error on the estimation bias when $\tau_{\max} = 1$: overestimation vs. underestimation.

Figure 8: Illustration of upper bounds and lower bounds on estimation bias in Theorem 2. (a) The case where the approximation errors of the Q-approximators are heterogeneous. (b),(d) The lower bound and the upper bound for given $\tau_1, \tau_2, A$. (c),(e) Due to time-varying feature of the approximation errors, the blue curve and the red curve depict the 'critical' points for the lower bound and the upper bound, respectively.

$$\geq \tau_{\max}(1 + \frac{1}{2^A(A+1)} + f_{AM} - 2g_{AM}) - \tau_{\min}(1 + \frac{1}{2^A})$$

$$\geq \tau_{\max}(1 + f_{AM} - 2g_{AM}) - \frac{3}{2}\tau_{\min}$$

$$\geq \tau_{\max}(1 + f_{AM} - 2g_{AM}) - 2\tau_{\min},$$

Summarizing, we obtain the following upper bound and lower bound on the estimation bias $\mathbf{E}[Z_M]$, in the general case with heterogeneous distributions for Q-function approximation errors:

$$\mathbf{E}[Z_M] \geq \gamma \left( \tau_{\min} - \tau_{\max}(f_{A(M-1)} + 2f_{AM}) \right);$$

$$\mathbf{E}[Z_M] \leq \gamma \left( 2\tau_{\min} - \tau_{\max} (f_{AM} - 2g_{AM}) \right).$$

□

## A.4 Numerical Illustration of Theorem 2

For a better understanding of Theorem 2, we next provide an example in Figure 8 following the same line as in Section 3.1. Figure 8(b) shows the case when $\tau_{\max}$ is small, i.e., $\tau_{\max} = 0.1$. Consistent with our analysis in Section 3.1, increasing the ensemble size $M > 10$ (red point) will lead to underestimation bias (upper bound is negative). It can be seen clearly in Figure 8(c) that, when $\tau_{\min}$ is small, the underestimation is the major issue when increasing the ensemble. On the other hand, Figure 8(d) demonstrates the case when $\tau_{\min}$ is large, i.e., $\tau_{\min} = \tau_{\max} = 1$. In this case, decreasing the ensemble size is 'likely' to cause overestimation. Similarly, we can observe from Figure 8(e) that when increasing $\tau_{\min}$, the ensemble size $M$ should be increased to avoid the overestimation (the blue shaded area). In this example, we set the number of actions as $A = 75$.

## A.5 Convergence Analysis of AdaEQ

Assume that at iteration $t$, the ensemble size is $M_t \leq N$, where $N$ is the number of approximators in the ensemble method. It is straightforward to verify that the stochastic approximation noise term has

Table 1: A comparison of hyperparameter settings among AdaEQ, REDQ [6] and AVG [2] implementation.

| Hyperparameter | AdaEQ (Our Method) | REDQ | AVG |
|---|---|---|---|
| Learning Rate | $3 \cdot 10^{-4}$ | $3 \cdot 10^{-4}$ | $3 \cdot 10^{-4}$ |
| Discount Factor ($\gamma$) | 0.99 | 0.99 | 0.99 |
| Optimizer | Adam | Adam | Adam |
| Target Smoothing Coefficient ($\rho$) | $5 \cdot 10^{-3}$ | $5 \cdot 10^{-3}$ | $5 \cdot 10^{-3}$ |
| Batch Size | 256 | 256 | 256 |
| Replay Buffer Size | $10^6$ | $10^6$ | $10^6$ |
| Non-linearity | ReLU | ReLU | ReLU |
| Number of Hidden Layers | 2 | 2 | 2 |
| Number of Hidden Unites per Layer | 256 | 256 | 256 |
| Number of Approximators ($N$) | 10 | 10 | 10 |
| Testing Trajectory Length $H$ | 500 | 500 | 500 |
| Initial Ensemble Size ($M_0$) | 4 | 2 | 10 |
| Ensemble Size Adaptation | True | False | False |
| Adaptation Frequency | Every 10 epochs | - | - |
| 'Tolerance' Parameter $c$ | 0.3 | - | - |

the contraction property as stated in [6] Appendix A.4 and [17] Appendix B. It follows that AdaEQ converges to the optimal Q-function with probability 1.

## B Experiments

### B.1 Hyperparameters and Implementation Details

In the empirical implementation, our code for AdaEQ is partly based on REDQ authors' open source code (`https://github.com/watchernyu/REDQ`) [6] and we use the identical hyperparameter setting, for the sake of fair comparison. For all three methods compared in our experiments, the first 5000 data points are obtained by randomly sampling from the action space without updating the Q-networks. For REDQ, we use the fine-tuned ensemble size $M = 2$ for all the MuJoCo benchmark tests. The results are similar with the reported results in the original paper. The detailed hyperparameter setting is summarized in Table 1.

### B.2 Additional Empirical Results on MuJoCo Benchmark

**Training Time Comparison.** In Table 2, we compare the average wall-clock training time over three training seeds. All the tasks are trained on the same 2080Ti GPU. It can be seen that the ensemble size adaptation mechanism does not significantly increase the training time for most tasks.

Table 2: A comparison of training time among AdaEQ, REDQ [6] and AVG [2] implementation.

| Environment | Hopper-v2 125K | Walker2d 300K | Ant 300 K | Humanoid 250K |
|---|---|---|---|---|
| AdaEQ | 62509.38 | 120598.13 | 130673.32 | 148786.28 |
| REDQ | 61565.28 | 118604.28 | 122954.04 | 104462.43 |
| AVG | 172115.94 | 151058.98 | 129526.58 | 124834.21 |
| AdaEQ/REDQ | 1.01$\times$ | 1.02$\times$ | 1.06$\times$ | 1.42$\times$ |

### B.3 Illustrative Examples for Time-Varying Approximation Errors

We present in Figure 9 an example to illustrate the time-varying nature of approximation errors during the training process. Following the setting in Section 2.2, we use 3 different approximators to approximate the true action-value. Each approximator is a 6-degree polynomial that fits the samples. The initial approximation errors used to generate samples are set to follow uniform distribution with

parameter $\tau = 0.3, 0.5, 0.7$, respectively. At each iteration, we first generate new samples from the approximator obtained in the last iteration and then update the approximator using these samples. The mean and standard derivation of the approximation error over different states are depicted in Figure 9(a) and 9(b). Clearly, the approximation error is time-varying and can change dramatically during the training process.

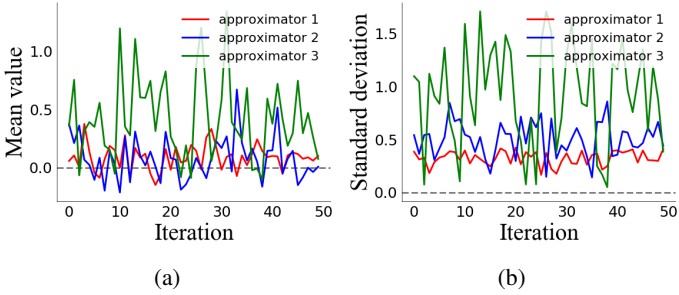

(a)                                        (b)

Figure 9: Illustration of the time-varying approximation error during the training process. (a) Mean approximation error over states. (b) Standard derivation of the approximation error over states.

## B.4 Performance Comparison with TD3 and SAC

Table 3: A Comparison of Average Return among AdaEQ (proposed method), REDQ [6], AVG [2], SAC [11] and TD3 [9]

| Environment | Hopper-v2 125K | Walker2d 300K | Ant 300K | Humanoid 250K |
|---|---|---|---|---|
| AdaEQ | 3372 | 4012 | 5241 | 4982 |
| REDQ | 3117 | 3871 | 5013 | 4521 |
| AVG | 1982 | 2736 | 2997 | 2015 |
| SAC | 2404 | 2556 | 2485 | 1523 |
| TD3 | 982 | 3624 | 2048 | - |