# OpenReview forum: "Adaptive Ensemble Q-learning: Minimizing Estimation Bias via Error Feedback"
_NeurIPS.cc/2021/Conference — NeurIPS 2021 Poster_

### Official Review · Reviewer_Cgf5 · 2021-07-16

**Rating:** 5
**Confidence:** 4

**Summary:**

This paper proposes an ensemble Q-learning which can adaptively adjust the ensemble size based on estimation error. The authors find that the ensemble min(Q_1, Q_2, ..., Q_M) may suffer from overestimation or underestimation with different M, so they propose to adapt M based on approximation error, which can minimize the estimation bias correspondingly.

The paper analyzes both in theoretically and empirically, show better performance than the existing ensemble q-learning methods on a few MuJoCo benchmark tasks.

**Main Review:**

The major contribution of this paper is to establish the correlation between the ensemble size and estimation bias under the ensemble q-learning framework(min operator).

Overall Clarity is good. The idea is clear and easy to follow, and the authors show the motivation why they propose this adaptive ensemble size q-learning.

Originality/Significance: I did not see exactly the same work before (to address the ensemble size). This paper tries to solve the ensemble size problem in ensemble q-learning, but it creates a new problem here, for example how to adjust the new hyperparameter c in Eq. 10. And the experiment show it impacts performance significantly in Fig.6(d). If we need to try different c and also need to set different M to get a good result, I do not see any practical advantage compared to TD3, SAC, etc

Regarding correctness of the algorithm: my main concern is when τ1 > τ2 > 0, \mathcal{K} \in [-τ1, τ1] but how do you get M\K \in [-τ2, τ2] in line 194-196. The problem is how do you define the set range \mathcal{M}. Another question your result in Eqs. 5 and 6 is from uniform distribution assumption, but you decide τ based on Gaussian in Eq. 9. Also you randomly sample in the following equation to get M does not make much sense to me.

Regarding experimental evaluation: The experimental results show this method is sensitive to new hyperparameter c and it needs to carefully tuned to get good result compared to REDQ. As the authors propose a new ensemble q-learning by adjusting the size M, it is important to compare with TD3 and SAC on a wide range of tasks in MuJoCo environments on which most current continuous control algorithms have been evaluated and compared.

Overall, the authors need to address these concerns above before acceptance.

**Time Spent Reviewing:**

20

---

> ### Author Response · Authors · 2021-08-10
> **Reply to Reviewer Cgf5 (2/2)**
>
> 3.[**Correctness of the algorithm**]
>
>
> - [How to obtain $\mathcal{M}$ and $\mathcal{K}$]
>
> Lines 194-196 considers an intuitive case of modeling Q-function approximation error. For instance, when only one approximator is chosen to be updated at each step [16], the approximation error distribution of this approximator would change over time and hence differ from the others.
> For instance, given $N$ approximators $Q^1, Q^2, \cdots, Q^N$, we can cluster those approximators into two groups by computing the standard derivation of $(Q^i – Q^{\pi})$ over a testing trajectory. $Q^{\pi}$ is the (discounted) MC return starting from the state-action pair in the testing trajectory. Under the uniform assumption of the approximation error, the empirical standard derivation is used to estimate $\tau_1$ and $\tau_2$ ($\tau_1 > \tau_2$). In order to obtain $\mathcal{K}$ and  $\mathcal{M}\setminus\mathcal{K}$, we then randomly sample from these two groups respectively.
>
> - [Rationale behind Eqn. (9)]
>
> We clarify that Eqn. (9) is not a Gaussian distribution. We use the empirical standard derivation to estimate $\tau_i$, and $\tau$ is the average value over all $\tau_i$:
> $$ \tau_i \approx \sqrt{3} std(Q^i- Q^{\pi}),  $$
> $$\tau \approx \frac{1}{ N} \sum_{i} \tau_i ,$$
> where we omit $\sqrt{3}$ in Eqn. (9) (by adjusting parameter $c$). We use $\tau$ to characterize the overall approximation error and further provide guidance to adjust the ensemble size. For example, as demonstrated in Theorem 2 and Figure 7 in Appendix B.2, when $\tau_{max}$ is small enough (thus the average $\tau$ is small, estimated in Eqn. (9)), increasing ensemble size is likely to cause underestimation; when $\tau_{\min}$ is large enough (the average $\tau$ is large), decreasing the ensemble size is likely to cause overestimation. We also give the detailed description in line 230 and Appendix B.2.
>
> - [Reason of using random sampling in Eqn. (9)]
>
> We use random sampling in Eqn. (10) for two reasons. (i) [*Noisy approximation error*]  We use Eqn. (9) to characterize the approximation error which is noisy in nature. In particular, Monte Carlo returns (with finite-length testing trajectory) may introduce empirical errors when estimating the underlying ground true value of $Q^{\pi}$. This noisy estimation is often the case when the policy is not deterministic, or the environment is not deterministic. Thus, we use random sampling to 'capture' the impact of this noisy estimation.
> (ii) [*Impact of ensemble size on estimation bias*]	Secondly, in general it is infeasible to characterize the exact relationship between  estimation bias $Z_M$ and ensemble size $M$ (see, e.g., Section 3 in ''Toward Provable Unbiased Temporal-difference Value Estimation'', NeurIPS'19). It is this challenge that motivated us to establish the upper bound and lower bound of the estimation bias (Theorem 1 and Theorem 2), based on which we  adapt the ensemble size, using random sampling,  in a particular range (increase in $[M_t+1, N]$ or decrease in $[2,M_t-1]$) to mitigate overestimation or underestimation. Without any further prior information (except from the bounds we obtained) about the approximation error, intuitively, the random sampling can be viewed as the 'exploration' in this RL algorithm.
>
>
>
> 4.[**Experimental evaluation**]
>
> In our work, we mainly compare AdaEQ with other ensemble based methods. Specifically, the comparison between REDQ and SAC has been reported in     Figures 1-3 in [6] and REDQ shows superior performance in a  large margin in the MuJoCo tasks Hopper, Walker, Ant and Humanoid. AdaEQ performs even better than REDQ in all the above four tasks. Besides, comparing with the average return of the same time steps reported by TD3 paper, AdaEQ performs better in Hopper (3.8x better, $1.2 \times 10^5$ time steps), Walker (1.24x better, $3 \times 10^5$ time steps) and Ant (2.36x better, $3 \times 10^5$ time steps) (Humanoid is not reported in the original TD3 paper).
>
> We will summarize in a table the performance comparison among AdaEQ, REDQ, AVG, SAC, TD3 on Hopper, Walker, Ant, Humanoid in our revision.

---

> > ### Comment · Reviewer_Cgf5 · 2021-09-01
> > **Thanks for the response**
> >
> > Thanks for your response on my comments. I think there is still some gap between algorithm (or theory) and experiments, so I will keep my rating.

---

> > > ### Author Response · Authors · 2021-09-01
> > > **Thank you for the feedback.**
> > >
> > > Thank you again for your response. Would you please elaborate the ‘gap’ between algorithm (or theorem) and experiments?
> > >
> > > In any case, we’d like to clarify further the connections among our algorithm, theory and experiments.
> > >
> > > - In our algorithm, there is no need to try different $c$. The parameter $c$ is pre-determined offline, by using the bounds obtained in Theorem 2 (please refer to bullet 1 in our earlier response). There is no need to try different $M$ since AdaEQ is designed to adapt the ensemble size $M$ during the learning process.
> > >
> > > - The  ensemble size is adapted by using random sampling as in bullet 3 in our earlier response.  In general, the noisy approximation error is inevitable and it is infeasible to characterize the exact relationship between estimation bias and ensemble size in deep learning.
> > >
> > > - In our algorithm, the approximation error is estimated under the same uniform assumption as in Theorem 2. Eqn. (9), not Gaussian distribution in your original comments (please refer to bullet 3 in our earlier response).
> > >
> > > - Our experiments are conducted using the algorithm described in Algorithm 1. It can drive the bias closer to zero compared with the previous ensemble methods, including TD3 and SAC (please refer to bullet 2 in our earlier response).

---

> > > > ### Comment · Reviewer_Cgf5 · 2021-09-11
> > > > **Thanks for your response**
> > > >
> > > > Thanks for your response. Overall, this paper tries to address the ensemble size M and my concerns:
> > > > (1) the complexity is linear with M, so it needs to verify its gain compared to AVG, TD3, SAC, etc
> > > > (2) if the hyperparameter $c$ can be determined offline by Theorem 2, why do you need to try different $c$ in experiments?
> > > > (3) when τ1 > τ2 > 0, $\mathcal{M}$ \ $\mathcal{K}$ is a subset of $\mathcal{K}$? Moreover, can you use uniform distribution to get τ˜_t in Eq. 9?
> > > > (4) Since M is randomly sampled, can you provide whether the bound in Theorem 1/2 is tight or not?

---

> > > > > ### Author Response · Authors · 2021-09-12
> > > > > **Thank you for the feedback**
> > > > >
> > > > > Thank you for your meticulous reading and useful feedback, which will help us to improve the presentation.
> > > > >
> > > > > (1) As stated in our earlier response, AdaEQ shows great practical advantages compared with AVG, TD3 and SAC. Firstly, it considers the time-varying characteristics of the approximation error due to the iterative nature of Q-learning. Secondly, AdaEQ simplifies the matter to fine-tune hyper-parameters (please refer to bullet 2 in our earlier response).
> > > > >
> > > > > Just to recap, the performance advantage of AdaEQ compared with AVG, TD3 and SAC can be summarized as follows: In our work, we compare AdaEQ with cutting-edge ensemble based methods,  including REDQ, Maxmin, and AVG. Specifically, the comparison between REDQ and SAC has already been reported in  Figures 1-3 in [6] and REDQ achieves superior performance over SAC by a large margin in the MuJoCo tasks (Hopper, Walker, Ant and Humanoid). AdaEQ performs even better than REDQ in all the above four tasks. Besides, comparing with the average return using the same time steps reported in TD3, AdaEQ performs better in Hopper (3.8x better, $1.2 \times 10^5$ time steps), Walker (1.24x better, $3 \times 10^5$ time steps) and Ant (2.36x better, $3 \times 10^5$ time steps) (Humanoid is not reported in the original TD3 paper).
> > > > >
> > > > > Per your comments, we will present the comparison in a table in the final version.
> > > > >
> > > > > (2) As stated before, there is no need to try different $c$. In our experimental studies, we use the same pre-determined $c$ for all the MuJoCo tasks in Section 4. We use different $c$ only in the Ablation Study in Figure 6(c) (d) to examine the performance sensitivity to $c$, since $c$ is a newly introduced parameter. It can be clearly seen that there is a wide range of choice for $c$ for AdaEQ to achieve good performance, indicating that AdaEQ is robust to $c$. This implies that even when the pre-determined $c$ offline is not optimized, it can still adequately serve the purpose (please refer to the last two paragraphs in bullet 1 in our earlier response).
> > > > >
> > > > > (3)  We would like to point out that $\mathcal{M}\setminus \mathcal{K}$ is not a subset of $\mathcal{K}$. For example, assume the approximation error of the 5 approximators $Q^1, Q^2, Q^3, Q^4, Q^5$ are $\tau = 1, 0.9, 0.5, 0.45, 0.4$ respectively. We obtain  $\mathcal{K}$ and $\mathcal{M}\setminus \mathcal{K}$ through a clustering process, namely,  we group the approximators with 'similar' approximation error together. Thus,  $Q^3, Q^4, Q^5 \in \mathcal{M}\setminus \mathcal{K}$ (e.g., $\tau_2 = 0.45$) and $Q^1, Q^2 \in \mathcal{K}$ (e.g., $\tau_1 = 0.95$).
> > > > >
> > > > > As stated before, Eqn. (9) is indeed derived by using the 'uniform distribution' instead of Gaussian distribution in your original comment. We use the empirical standard deviation to estimate uniform distribution parameter  $\tau_i$, and $\tau_t$ is the average value over all $\tau_i$ (please refer to the detailed response in bullet 3 in our earlier response).
> > > > >
> > > > > (4)  We note that the upper bound obtained in Theorem 1 (for the case with two distributions for approximation errors) is tight whereas the lower bound is not. In particular, let $K \rightarrow 0$ and $\tau_1 \rightarrow \tau_2$, the upper bound in Theorem 1 is $\gamma \tau_1 [1-2f_{AM}]$, which encompasses the results obtained in Theorem 1 [16] of the Maxmin paper, where the equality is achieved.
> > > > >
> > > > > To the best of our knowledge, our work is the first to deliver both bounds under the general assumption. The previous work REDQ and Maxmin either considered a simpler case with the same distribution or didn't provide the bounds on the estimation bias.
> > > > >
> > > > > As noted in Ln 223-225, the bounds in Theorem 2 (for the general heterogeneous case) is not as sharp as Theorem 1 since the order statistics over heterogeneous distributions is technically more challenging (e.g., PK Sen. ''A note on order statistics for heterogeneous distributions.'' The Annals of Mathematical Statistics'1970).

---

> ### Author Response · Authors · 2021-08-10
> **Reply to Reviewer Cgf5 (1/2)**
>
> Thank you for your meticulous reading and valuable comments.
>
>
>
> 1.[**How to adjust tolerance parameter $c$**]
>
> The main purpose of our work is to reduce the estimation bias to close to zero by adaptively changing ensemble size during the learning process. Guided by the upper bound and lower bound in Theorems 1 and 2, we introduce a tolerance parameter $c$ in AdaEQ to enable the ensemble size adaptation to drive the estimation bias to be close to zero; as a result the estimation bias may dynamically oscillate between positive and negative.  Based on  Theorems 1 and 2, the tolerance parameter $c$ can be determined as follows.
>
>
> - [Step 1:  Estimation of $ \tau_{\min}$ and $\tau_{\max}$] We run AdaEQ for a few epochs  with a fixed ensemble size to obtain the estimation of the approximation error distribution parameter $ \tau_{\min}$ and $\tau_{\max}$.   Specifically, $\tau_{\min}$ and $\tau_{\max}$ can be estimated by using the method introduced in lines 251-258 using testing trajectories over the few epochs. We run a testing trajectory from a random initial state using the current policy to compute the (discounted) MC return $Q^{\pi}$ and the estimated Q-function value $Q^i, i=1,2, \cdots, N$. Then we fit a uniform distribution model $\mathcal{U}(-\tau_i,\tau_i)$ of the approximation error $(Q^i-Q^{\pi})$ for each Q-function approximator $Q^i$. It follows that the $\tau_{\min}$ and $\tau_{\max}$ can be obtained by choosing the minimum and maximum value among $\tau_i, i =1, 2, \cdots, N$.
>
> - [Step 2:  Bounds on estimation bias]  The upper bound and the lower bound in Theorems 1 and 2 can be computed by using $\\{ \tau_{\min}, \tau_{\max}, A, \gamma\\}$. Then we investigate the relationship between ensemble size $M$ and the estimation bias  by studying the bounds (e.g., Figure 4(b)). Next, we can identify the 'critical' points as illustrated in Figure 4(b), where the red  point is found by setting the upper bound  to zero and the blue point is obtained by setting the lower bound  to zero.  Observe that a 'proper' ensemble size should be chosen between the 'critical' points, so to reduce the overestimation and underestimation bias as much as possible. Since the approximation error is time-varying during the learning process,   these two 'critical' points change along with $\{\tau_{\max}\}$ and $\{\tau_{\min}\}$ (as shown in Figure 4(c)). Clearly,  it is desirable to drive the system to avoid the red region (underestimation) and the blue region (overestimation).
>
> Next, we have a few remarks on AdaEQ's robustness to parameter  $c$.  As illustrated in Figure 4(c),  when AdaEQ stays in the white region, AdaEQ can work well in mitigating both underestimation and overestimation. It can be clearly seen that there is a wide range of choice for parameter $c$ (e.g., $[0.5,0.7]$ in Figure 4(c)) for AdaEQ to stay in the white region, indicating that AdaEQ is not sensitive to $c$. This indicates that even though the pre-determined $c$ in steps 1 and 2 above is not optimized, it can still serve the purpose.
> In particular, for a given $c$, when the current approximation error parameter $\tau_{\max} > c$, the overestimation becomes more pronounced, and we can increase the ensemble size to reduce overestimation and drive AdaEQ to be in the white region; and when the current approximation error parameter $\tau_{\max} < c$, we should pay more attention to the underestimation and we can decrease the ensemble size to address it. The concrete example of choosing $c$ is also given in lines 212-218.
>
>
>
> Further, our experimental results corroborate that AdaEQ is not sensitive to parameter $c$ in MuJoCo environment. (i) [*Robustness across different tasks*] In Hopper, Ant, Walker (Figure 4), Humanoid (Figure 8 in Appendix B.2), we use the same parameter $c = 0.3$ in all the MuJoCo tasks. The results show that the same value $c=0.3$ works well to reduce the estimation error comparing with other methods. (ii)[*Robustness to $c$ in a wide range*]  In Figure 6(d), we compare the estimation bias over a wide range  of parameter $c$ in Hopper-v2 task. The estimation bias is consistently closer to zero when c is in range [0.3, 0.7] comparing with REDQ or AVG. (iii) [*Degeneration case of AdaEQ*] In Figure 6(c), the performance of AdaEQ  degrades only when using 'extreme’ values for parameter $c$. This phenomenon is summarized in lines 271-274. In this way, AdaEQ can be viewed as a generalization of Maxmin and REDQ with ensemble size adaptation. Intuitively, when $c$ is set to be 'extreme’ large (e.g., $c=1.5$), AdaEQ  degenerates to REDQ with ensemble size 2; in contrast, when $c$ is set to be ‘extreme’ small (close to zero, e.g., $c=0.001$), AdaEQ  degenerates to Maxmin with ensemble size $N$. Both methods cannot  drive the estimation bias to zero during the learning process, leading to  either the overestimation or underestimation.
>
> 2.[**Practical advantage of AdaEQ compared with TD3 and SAC**]
>
> Given the time-varying nature of the approximation error, a fixed ensemble size, used in previous ensemble methods, would result in undesirable overestimation or underestimation. AdaEQ adapts the ensemble size $M$ on the fly to drive the estimation bias to be close to zero and achieve better performance. It does not try different ensemble size $M$ to get a good result. As noted above, AdaEQ is robust to a wide range of choice of parameter $c$. Clearly, AdaEQ can achieve better performance compared to previous ensemble methods by driving the estimation bias close to 0, without increasing the algorithm complexity: we introduce the parameter $c$ to adaptively adjust the ensemble size, but eliminate the need of fine-tuning the initial ensemble size before the learning process.
>
> More specifically, the algorithm complexity of AdaEQ can be understood from the following two aspects: (1) [*Robust to $M$*] In the previous ensemble methods, $M$ is a hyperparameter and need the fine-tuning. The results [6, 16] show that the performance is very sensitive to the choice of $M$ (e.g., in Figure 2(d), changing ensemble size from 2 to 3 will lead totally opposite estimation bias). In stark contrast, the performance of  AdaEQ is much more robust to the initial selection of $M$ (Figure 6(a), 6(b)).  (2) [*Less  sensitive to $c$*] AdaEQ is not  sensitive to the parameter $c$ in a wide range (see, e.g., our response on how to choose parameter $c$ above).
>
>
> - [Comparison with SAC and TD3]
>
> The fact that the approximation error is time-varying, due to the iterative nature of Q-learning, has not been taken into account in  SAC and TD3. Consequently, the estimation bias is either positive or negative after executing the algorithm like SAC or TD3 since there is no further ‘control’ mechanism during the learning process. Meantime, in practice, it is unclear a priori whether those algorithms would lead to overestimation or underestimation bias in different tasks once the hyperparameters are determined for tasks. For example, recent works identify that TD3 can suffer from large underestimation bias, which also impacts performance (Ciosek et al. ''Better exploration with optimistic actor critic'' NuerIPS'19). SAC suffers from cumulative estimation bias and does not work well for tasks  in high-dimensional action space  (e.g., Ant, Humanoid, Walker [6]). Thus motivated,  AdaEQ  aims to drive the bias close to zero based on the underlying learning dynamics, by introducing the 'Model Identification Adaptive Control (MIAC)' [3, 24] mechanism during the learning process.
>
> Due to the complexity of the design space in deep RL, fine-tuning hyperparameters (e.g., clipping noise parameter $c$ in TD3 and SAC, learning rate in deep RL)  or finding an 'optimal' parameter setting is a crucial step to guarantee the performance. For example, recent work proposes SD2 (a TD3 type of algorithm), where the same clipping trick is applied. It has shown that clipping parameter $c$ has impact on estimation bias and the algorithm performance, e.g., Pan et al. ''Softmax deep double deterministic policy gradients'' NeurIPS’20. (Theorem 3: ''...there exists noise clipping parameter $c>0$  that bias(SD2)$\leq $bias(DDPG)'')
>
> In a nutshell, AdaEQ does not need to fine-tune initial ensemble size $M$ to get a good result. Further, AdaEQ is also robust to the choice of parameter $c$ in a wide range, thereby simplifying the matter to fine-tune hyper-parameters.

---

### Official Review · Reviewer_u8Dd · 2021-07-16

**Rating:** 4
**Confidence:** 4

**Summary:**

This paper proposes an adaptive ensemble size to prevent the over/under estimation bias that a too small/large ensemble creates due to time-varying approximation errors. They derive bounds on the estimation bias for an argmin ensemble method and propose an adaptive ensemble method to mitigate this over/under estimation bias problem.

**Limitations And Societal Impact:**

Limitations: Addressed
Societal Impact: Not relevant

**Main Review:**

Overall: The paper proposes an interesting problem and an interesting solution. However the paper
spends a lot of time using simple examples to show intuition where it is not clear whether these examples are “degenerate” examples (Figures 2 and 3 are unnecessary and so is Theorem 1 which is a simple case of Theorem 2). Furthermore they need to spend to spend more time justifying their more unusual generative assumptions (Uniform noise and using the argmin ensemble estimator). It is not clear whether this bias changing “signs” above a certain ensemble size problem is an artifact of the unusual “min” estimator.

Major Comments:

Line 122: Is the overestimation bias being positive proven in [16] or [6] or somewhere else, if so please state where and how. Otherwise, this seems like an extremely aggressive assumption. Or if by definition a bias being positive implies overestimation, then why can the bias not be negative? Is this not underestimation?

Why is it assumed that the approximation error has Uniform noise, not Gaussian noise? Furthermore the variance of distribution of the noise for each of the ensemble members depends heavily on \tau. For example if \tau is large, then the N \tau_i have higher variance, and this in turn makes the distributions of each error_i be more different from each other.

If too large of an ensemble size causes underestimation bias, and this bias is negative according to Line 189, why does averaging not reduce bias overall. Furthermore, why is Z_M being studied for only the argmin ensemble method and not the averaging ensemble method in Line 188/189. If only the argmin estimator exhibits both positive and negative biases, wouldn’t averaging an “ensemble” of those estimators create an unbiased estimator?

What is K and what is M in the example described after Theorem 1?

In the experiments in Figure 5, why is the bias almost always positive even for the “optimized” methods? Shouldn’t the bias be as likely to be positive as negative if the methods have been “optimized” and centered? Also what is the very large dip (between 1 and 1.5) in the AdaEQ model for Ant?

Minor Comments:

Line 106 missing “the” in “By definition, (the) Q-function is the expected return when” and this phrase is strangely worded “and following with the policy”

Line 230: Typo “thew”


**Time Spent Reviewing:**

3

---

> ### Author Response · Authors · 2021-08-10
> **Reply to Reviewer u8Dd (2/2)**
>
> 2.[**Assumption on Approximation Error Distribution**]
>
> The uniform distribution assumption for the Q-function approximation error has been widely used in the existing studies (see, e.g., REDQ [6], Maxmin [16]), and this is one main reason we impose this assumption in this paper. The uniform distribution assumption for the function approximation error can be traced back to Thrun et al.’s work in 1993 [27] and has also been used in the following recent papers:
>
> - Van Hasselt et al. ''Deep reinforcement learning with double q-learning.'' AAAI’16 (where in Appendix Theorem 2, they prove an upper bound of estimation bias under the uniform distribution assumption)
>
> - Anschel et al. ''Averaged-DQN: Variance reduction and stabilization for deep reinforcement learning.''  ICML’17
>
> - Wu et al. ''Reducing estimation bias via triplet-average deep deterministic policy gradient.'' IEEE TNNLS’20
>
> - Lee et al. ''Choice of approximator and design of penalty function for an approximate dynamic programming based control approach.'' IFAC’06
>
>
> In fact, modeling approximation error is an open problem, because the optimal Q-function is generally unknown and the approximation error for different tasks varies. Without any prior information about the approximation error, it makes sense to use a uniform distribution to indicate that both positive and negative approximation error are possible in Q-function approximators.
>
>
> We would like to point out that from a theoretic perspective, it is possible to use the powerful machinery of Order Statistics to characterize the relationship between ensemble size $M$ and estimation bias $\mathbf{E}[Z_M]$ under a general distribution of the approximation error, but explicit bounds on the estimation bias may be not attainable. Specifically, in the general case discussed in Theorem 2,
>
> Assume that the approximation error $e_i(s,a)$ for each Q approximator $Q^i(s,a), i=1,2,\cdots,M$ is with PDF $f_i(x)$ and  CDF $F_i(x)$. From order statistics (see Appendix A.1 Lemma 1), we have the CDF and PDF for $X_{1:M} = \min_{i} e_i$ as follows,
>
> $$    F_{1:M}(x) =\mathbf{P} (X_{1:M} \leq x) = 1- \prod_{i=1}^{M}(1-F_i(x)), $$
>  $$   f_{1:M}(x) =\sum_{i=1}^M \left( f_i(x)\prod_{j\neq i}\left(1-F_j(x)\right) \right), $$
>
> Assume that at state $s'$, there are $A$ actions applicable. Then the estimation bias $Z_{M}$ is
> $$
> \mathbf{E}[Z_{M}]
> = \gamma \mathbf{E}[\max_{a'} \min_{i\in \mathcal{M}} e^i(s,a)]
> =\gamma\left[ \int_{lower}^{upper} Axf_{1:M}(x)F_{1:M}(x)^{A-1}dx \right].
> $$
>
> As expected, it is highly non-trivial, if not impossible, to derive the explicit bounds of estimation bias when the CDF and PDF is given as in the general form.
>
> In a nutshell, in deriving Theorem 1 and 2 we follow the the existing literature to impose the uniform distribution assumption for the Q-function approximation error . To the best of our knowledge, our work is the first to deliver both bounds under this assumption. The full proof can be found in Appendix A.1 and A.3.
>
> [**Variance of distribution parameter $\tau_i$**]
>
> Our work does not investigate the variance of the distribution parameter $\tau_i$.
> We are not exactly sure about your comment on ''…$N \tau_i$ have higher variance, this in turn makes the distribution of each error be more different''.
>
> We would like to point out that the general case with heterogeneous distribution for Q-function approximation errors is addressed in Line 219 and Theorem 2. We consider that the distribution for each approximator’s approximation error can be different and change over time. In this case, the derived upper bound and lower bound in Equation (7)(8) (illustration is in Appendix A.4) reveals that: when $\tau_{max}$ is small enough (thus the average estimation error parameter $\tau \approx \frac{1}{N} \sum_i\tau_i$ is small, estimated in Equation (9)), increasing the ensemble size is likely to cause underestimation; when $\tau_{\min}$ is large enough (the average $\tau$ is large), decreasing the ensemble size is likely to cause overestimation. We give the detailed description in Line 230 and Appendix B.2.
>
>
> 3.[**Why we use $\min$ operator**]
>
> Our work aims to reduce estimation bias to close to zero, rather than keep the positive or negative bias, since both biases can degrade the performance [16]. As we noted in Line 121, taking average over all positive (negative) bias still results in positive (negative) bias [see, e.g., Chen et al. ''Randomized ensembled double q- learning'', ICLR'21]
>
>
>
> We are not sure if we understand  your comment ''if only the argmin estimator exhibits both positive and negative biases, wouldn't averaging an 'ensemble' of those estimators create an unbiased estimator?'' Would you please clarify your question?
>
> It is worth noting that the estimation bias is measured by $\mathbf{E}[Z_M]$ (Line 189), which is deterministic and has only one 'sign', i.e., positive or negative (or zero),   once the approximation error model and the ensemble size is given. Our work exactly utilizes the relationship between the ensemble size and estimation bias (as captured by the bounds in Theorem 1 and Theorem 2) to adapt ensemble size during the learning process, such that the cumulative estimation bias is minimized.
>
> At the cost of higher complexity, it is possible to construct two proxy estimators (using Eqn. (3), Line 123) with two non-overlapping subsets of Q-function approximators, each with different ensemble size, and hopefully one induces positive bias and the other negative bias, respectively, i.e., $ Q^{prox1}=\min_{i\in M_1} Q^i $,  $Q^{prox2}=\min_{i\in {M}_2}Q^i $. However, even with this very complicated approach,
> it is highly unlikely  to create an exactly 'unbiased' estimator by  averaging the positive biased estimator and the negative biased estimator in deep reinforcement learning, where the underlying true value is generally unknown.
>
> Moreover,  it has been shown that the average ensemble method performs worse than the ensemble methods using $\min$ operator in both discrete action space (e.g., Asterix, Pixelcopter, Breakout [16]) and continuous action space (e.g., Hopper, Ant, Walker [6]). Our experiments in Section 4 also give the same conclusion. The analysis on the estimation bias with average operator has been conducted in Average-DQN paper [2]. Our analysis focuses on the ensemble methods with a $\min$ operator since it is promising to deliver  better performance.
>
> Different from the averaging method, we are able to bring the estimation bias from positive to negative by adjusting the ensemble size. This is not a disadvantage but an advantage of AdaEQ with $\min$ operator. We utilize the relationship between estimation bias and the ensemble size (specifically, the bounds in Theorems 1 and 2) to adapt the ensemble size during the learning process to make the estimation bias oscillate between positive and negative, in order to minimize the accumulated estimation bias over the learning process.
>
> 4.[**Definition of $M$ and $K$**]
>
> The meaning of $K$ and $M$ in Theorem 1 is stated right before Theorem 1, please refer to line 194. We have introduced $K$ and $M$ also in Section 2 and Figure 4(a). $K$ is the size of the set where the approximation error follows distribution 1 and $M$ is the total ensemble size.
>
> 5.[**Why the curve is not 'centered' in Figure 5**]
>
> The estimation curve in Figure 5 is not exactly centered for two reasons. (i) We use Eqn. (9) to characterize the current approximation error which is noisy in nature. In particular, Monte Carlo returns (with limited length of testing trajectory) may introduce empirical errors when estimating the underlying ground true value of $Q^{\pi}$. This estimation is noisy when the policy is not deterministic, or the environment is not deterministic. (ii) Secondly, in general it is infeasible to characterize the exact relationship between  estimation bias $Z_M$ and ensemble size $M$  such that the unbiased Q-value can be extracted easily (the curve in Figure 5 will then be centered to zero in this case) (see, e.g., Section 3 in ''Toward Provable Unbiased Temporal-difference Value Estimation'', NeurIPS'19).  Thus motivated, we establish the upper bound and lower bound of the estimation bias (Theorems 1 and 2), based on which, we are able to adjust the ensemble size in the corresponding range (increase in $[M_t+1, N]$ or decrease in $[2,M_t-1]$) to overcome  overestimation or underestimation. As illustrated in Figure 5, comparing with previous ensemble methods, AdaEQ can indeed further reduce the bias closer to zero. Notably in Ant and Walker-2d, the bias is mostly centered around zero. The similar result of the estimation bias curve of the other two methods is presented in REDQ paper [6] where the bias is constantly positive during the training process even with an 'optimal' choice of ensemble size.
>
> [**The dip in the Ant Experiments**]
>
> The large dip in Ant environment is also observed in REDQ paper (see Figure 3(e) therein). The instability during training often originates from the poor policy. Intuitively, when the ensemble size is too large or too small, the estimation error accumulates quikcly in the continuous action space and hence leads to a poor policy. AdaEQ is designed to overcome this issue when a 'bad’ initial ensemble size is given. In our experiments, the initial ensemble size for AdaEQ is not 'optimal’ (e.g., obtained by grid-searching in REDQ), which yields poor performance in the early stage. However, it can be clearly observed that AdaEQ is able to reduce the estimation bias close to zero during the following learning process.
>
> Our response to the minor comments: we will modify the two typos in our revision.

---

> ### Author Response · Authors · 2021-08-10
> **Reply to Reviewer u8Dd (1/2)**
>
> We appreciate your useful review and suggestions.
>
> First, here is our response to the overall comments:
>
> 1.[**Illustrative Example in Section 2.2**]
>
> Due to space limitation, we have relegated the demonstration of Theorem 2 to Appendix A.4, which presents the detailed illustration. The illustration in Figure 2 is in the same spirit as in the well-received Double DQN paper (van Hasselt et al., AAAI'16) [31]. The fitting process is to demonstrate the function approximation in DQN. The ensemble process strictly follows the ensemble method as described in Section 2.1. The estimation error is also evaluated using the same metric as we used in Section 5.
>
> We submit that this illustration in Figure 2 is essential to motivate the study and help the reader to understand the problem more clearly. In previous work, the impact of the ensemble size selection on the approximation error  is not well understood. In particular,  all prior studies use a fixed ensemble size, but the fact that the approximation error is time-varying, due to the iterative nature of Q-learning, gives rise to the fundamental question  whether a fixed ensemble size (e.g. REDQ, AVG, Maxmin) should be used in the learning process. Thus,  Figure 2 and Figure 3 are used to reveal that a fixed ensemble size can lead to consistent overestimation and underestimation which would degrade the cumulative reward  significantly.
>
> This has not been shown in prior arts. Figure 3 clearly demonstrates that the adaptive ensemble size is able to drive the bias close to zero, which motives our analysis in Theorem 1, 2 and the development of AdaEQ.
>
> 2.[**Theorems 1 and 2**]
>
> We would like to emphasize that Theorem 1 is not a degenerated case of Theorem 2.  Theorem 1 provides a sharper bound on the estimation bias in the case  approximation errors following one of the two distributions, i.e., $
> \mathbf{E}[Z_{M}] \geq \gamma\left(\tau_1 (1-2f_{AK}-2f_{AM}) + \tau_2(1-f_{AM}-(1-\beta_K)^A)\right)$;  $ \mathbf{E}[Z_{M}] \leq \gamma\left(\tau_1 +\tau_2(1- 2f_{A(M-K)}-(1-\beta_K)^A)\right)$. Consequently, as illustrated in Figure 4(c), the gap between underestimation (red shaded area) and overestimation (blue shaded area) is smaller.  On the contrary, Theorem 2 studies the general case with heterogeneous error distributions across all Q-function approximators; as expected, this case is technically more challenging and the bounds would not as sharp as in Theorem 1. In fact, the gap between underestimation (red shaded area) and overestimation (blue shaded area) is wider (Figure 7(c)in Appendix A.4).
>
> 3.[**Why we use $\min$ Operator**]
>
> We are not sure about your comment ''It is not clear whether this bias changing 'signs' above a certain ensemble size problem is an artifact of the unusual 'min' estimator ''.
>
> First, applying $\min$ operator is a widely used technique to address the overestimation problem in Q-learning, e.g.,  it has been  used in the following works:
>
> - Fujimoto et al. ''Addressing Function Approximation Error in Actor-Critic Methods'', ICML'18.
>      In the proposed TD3 algorithm, the $\min$ is used in the  target $y$: $y = r + \min_{i=1,2}Q_i(s',a)$
>
> - Kumar et al. ''Stabilizing off-policy q-learning via bootstrapping error reduction'', NeurIPS’19
>
> - Kuznetsov et al. ''Controlling overestimation bias with truncated mixture of continuous distributional quantile critics.'' ICML’20
>
> - Pan et al. ''Softmax deep double deterministic policy gradients'' NeurIPS’20
>
> - Shao et al. ''GRAC: Self-Guided and Self-Regularized Actor-Critic'' NeurIPS’20
>
> It is well-known in RL that the classic Q-learning method suffers from the overestimation problem (we include the detailed explanation of overestimation problem below). To address this overestimation problem (positive bias), $\min$ operator in the Bellman backup is introduced in  ensemble methods so as to reduce the estimation bias. However,
> using a fixed ensemble size during the learning process could lead underestimation (negative bias) when the ensemble size is large, which is  not desirable either. This 'changing sign' phenomenon is not an artifact problem but a  drawback of previous ensemble methods for using a fixed ensemble size. In fact,
> motivated by this 'changing sign' phenomenon, we propose AdaEQ to adaptively change ensemble size during the learning process to drive the estimation bias to be close to zero (which may fluctuate  between positive and negative), such that the accumulated estimation bias over the learning process is minimized.
>
> Moreover, the bias 'changing signs' is not a unique phenomenon  for '$\min$' operator. For example, Double Q-learning (NeurIPS’10) [11] uses two independent estimators to make target value estimation but doesn't use the $\min$ operator. It also suffers from underestimation during the learning process under certain problem settings (Lemma 1 [11], [6, 16]).
>
> In what follows, we provide point-to-point responses to your comments.
>
> 1.[**Overestimation Problem in Q-learning**]
>
> By definition, overestimation in Q-learning gives a positive bias in the target (as the word over-estimation indicates). It is not an assumption, but an inherent problem caused by the $\max$ operator in classic Q-learning, which was first identified in Thrun et al.’s work in 1993 [27]. Recent works, e.g., Double Q-learning, Double DQN, TD3 and ensemble methods [2, 6, 16, 18] have all studied this problem. More specifically, the overestimation problem can be described as follows [11]:
>
> In the Bellman operator, classic Q-learning uses a single estimator to estimate the value of the next state: $\max_a Q(s’,a)$ is an estimate for $\mathbf{E}[\max_a Q(s’,a)]$ which in turn approximates $\max_a \mathbf{E}[Q(s’,a)]$.
> According to Jensen’s Inequality, we have $\mathbf{E}[\max_a Q(s’,a)] \geq \max_a \mathbf{E}[Q(s’,a)]$, which indicates that in Bellman backup Equation (2), the estimated target value tends to overestimate the underlying true value.
> Empirically, the overestimation issue has also been widely observed [8, 11, 31].
>
> The estimation bias can be negative (underestimation) when applying certain algorithms (e.g. Double Q-learning) in some tasks, as also illustrated in Figure 2 and Figure 3 in this paper.

---

### Official Review · Reviewer_54UB · 2021-07-17

**Rating:** 6
**Confidence:** 3

**Summary:**

To mitigate the overestimation issue in Q-learning, this paper proposes an adaptive way to control the ensemble size of Q-function approximators.
Experimentally, this paper demonstrates how the ensemble size affects the estimation bias.
Theoretically, this paper gives the upper bound and lower bound of the estimation bias according to the ensemble size.
Finally, this paper evaluates the proposed algorithm in the MuJoCo environment.


**Limitations And Societal Impact:**

(1) This paper states in line 264-265 that "Recall that parameter c can be determined by using the upper bound and the lower bound in Theorem 2 (Theorem 1)." However, the upper bound and lower bound are determined by \tau_\min and \tau_\max, which are unknown variables.
It is not clear how to determine c by the upper and lower bound, since upper and lower bound are unknown.
(2) This paper studies the estimation bias of the ensemble size. It lacks analysis of how ensemble size affects the cumulative reward.
(3) This paper selects three MuJoCo environments to evaluate the proposed algorithms. How about the rest MuJoCo environments? More experiments on discrete control tasks can make more evaluation of the algorithm.


**Main Review:**

Originality:
This paper studies the effect of ensemble size on the estimation bias. There are several works studying the ensemble size of Q-function approximators. But this is the first to study how to adaptively control ensemble size on the fly. The originality is incremental.

Quality:
In response to the estimation bias, this paper discusses the upper and lower bounds experimentally and theoretically.

Clarity:
This paper is well written, clear and easy to read.

Significance:
The ensemble size heavily affects the estimation bias. This paper proposes a randomized algorithm which updates the ensemble size according to a pre-determined "tolerance". It is a rough way to adjust the ensemble size. In total, this paper makes an incremental contribution.


**Time Spent Reviewing:**

3

---

> ### Author Response · Authors · 2021-08-10
> **Reply to Reviewer 54UB**
>
> Thank you for your thorough review and constructive feedback.
>
> (1) [**Determination of parameter $c$ and AdaEQ's robustness to  $c$**]
>
> First, we outline the steps to determine the tolerance parameter $c$ as follows:
>
> - [Step 1:  Estimatation of $ \tau_{\min}$ and $\tau_{\max}$] We run AdaEQ for a few epochs  with a fixed ensemble size to obtain the estimation of the approximation error distribution parameter $ \tau_{\min}$ and $\tau_{\max}$.   Specifically, $\tau_{\min}$ and $\tau_{\max}$ can be estimated by using the method introduced in lines 251-258 using testing trajectories over the few epochs. We run a testing trajectory from a random initial state using the current policy to compute the (discounted) MC return $Q^{\pi}$ and the estimated Q-function value $Q^i, i=1,2, \cdots, N$. Then we fit a uniform distribution model $\mathcal{U}(-\tau_i,\tau_i)$ of the approximation error $(Q^i-Q^{\pi})$ for each Q-function approximator $Q^i$. It follows that the $\tau_{\min}$ and $\tau_{\max}$ can be obtained by choosing the minimum and maximum value among $\tau_i, i =1, 2, \cdots, N$.
>
> - [Step 2:  Bounds on estimation bias]  The upper bound and the lower bound in Theorems 1 and 2 can be computed by using $\\{\tau_{\min}, \tau_{\max}, A, \gamma \\}$. Then we investigate the relationship between ensemble size $M$ and the estimation bias  by studying the bounds (e.g., Figure 4(b)). Next, we can identify the 'critical' points as illustrated in Figure 4(b), where the red  point is found by setting the upper bound  to zero and the blue point is obtained by setting the lower bound  to zero.  Observe that a 'proper' ensemble size should be chosen between the 'critical' points, so to reduce the overestimation and underestimation bias as much as possible. Since the approximation error is time-varying during the learning process,   these two 'critical' points change along with $\{\tau_{\max}\}$ and $\{\tau_{\min}\}$ (as shown in Figure 4(c)). Clearly,  it is desirable to drive the system to avoid the red region (underestimation) and the blue region (overestimation).
>
> Next, we have a few remarks on AdaEQ's robustness to parameter  $c$.  As illustrated in Figure 4(c),  when AdaEQ stays in the white region, AdaEQ can work well in mitigating both underestimation and overestimation. It can be clearly seen that there is a wide range of choice for parameter $c$ (e.g., $[0.5,0.7]$ in Figure 4(c)) for AdaEQ to stay in the white region, indicating that AdaEQ is not sensitive to $c$. This implies that even though the pre-determined $c$ in steps 1 and 2 above is not optimized, it can still serve the purpose.
> In particular, for a given $c$, when the current approximation error parameter $\tau_{\max} > c$, the overestimation becomes more pronounced, and we can increase the ensemble size to reduce overestimation and drive AdaEQ to be in the white region; and when the current approximation error parameter $\tau_{\max} < c$, we should pay more attention to the underestimation and we can decrease the ensemble size to address it. The concrete example of choosing $c$ is also given in lines 212-218.
>
> Further, our experimental results corroborate that AdaEQ is not sensitive to parameter $c$ in the MuJoCo environment.  (i) [*Robustness across different tasks*] In Hopper, Ant, Walker (Figure 4), Humanoid (Appendix B.2 Figure 8), we use the same parameter $c = 0.3$ in all the MuJoCo tasks (Figure 5 and Appendix B.2 Figure 8). The results shows that the same value $c=0.3$ works more effectively to reduce the estimation error comparing with other methods. (ii) [*Robustness to parameter $c$ in a wide range*] In Figure 6(d), we compare the estimation bias over a wide range (from 0.001 to 1.5) of parameter $c$ in Hopper-v2 task. The estimation bias is consistently closer to zero when c is in range $[0.3, 0.7]$ comparing with REDQ or AVG. (iii) [*Degeneration case of AdaEQ*] In Figure 6(c), the performance of AdaEQ  degrades only when using 'extreme’ values for parameter $c$. This phenomenon is summarized in lines 271-274. In this way, AdaEQ can be viewed as a generalization of Maxmin and REDQ with ensemble size adaptation. Intuitively, when $c$ is set to be 'extreme’ large (e.g., $c=1.5$), AdaEQ  degenerates to REDQ with ensemble size 2; in contrast, when $c$ is set to be ‘extreme’ small (close to zero, e.g., $c=0.001$), AdaEQ  degenerates to Maxmin with ensemble size $N$. Both REDQ and Maxmin methods cannot  drive the estimation bias to zero during the learning process, leading to  either the overestimation or underestimation.
>
> (2)[**Relationship between cumulative reward and the ensemble size $M$**]
>
> As is standard, in Q-learning, the (discounted) cumulative reward, namely the current Q-function value $Q^i(s,a)$, is estimated by $M$ (ensemble size) Q-function approximators as follows.
>
> Given a transition sample $(s,a,r,s’)$, the Bellman operator can be used to update the Q-function in AdaEQ as (Equation (2), line 111):
>
> $$Q^i(s,a) \leftarrow (1-\alpha)Q^i(s,a) + \alpha y, \quad y := r + \gamma \max_{a'\in \mathcal{A}}\min_{j\in\mathcal{M}}Q^j(s',a'),$$
>
> where $y$ is the target value calculated by using $Q^j, j=1,\cdots,M$. The  optimal target value is obtained by using optimal Q function and denoted as $y^{\*}$. Recall that $\mathbf{E}[Z_M] := \mathbf{E}[y-y^{*}]$, then we have,
>
> $$ \mathbf{E}[Q^i(s,a)] \leftarrow (1-\alpha)\mathbf{E}[Q^i(s,a)] + \alpha y^{*} + \alpha \mathbf{E}[Z_M].$$
>
> Thus, at the $n$-th iteration, the relationship between (estimated) cumulative reward and the optimal reward is as follows:
>
> $$ \mathbf{E}[Q^i(s,a)] = (1-\alpha)^n\mathbf{E}[Q^i(s,a)] + (1 - (1-\alpha)^n) y^{*} + \sum_{t=0}^{n}\alpha(1-\alpha)^t \mathbf{E}[Z_{M_t}],$$
> where $M_t$ is the ensemble size at iteration $t$.
>
>
> From which we can observe that as $n \to \infty$, $ \mathbf{E}[Q^i(s,a)] \to  y^{*} + \sum_t\alpha(1-\alpha)^t \mathbf{E}[Z_{M_t}] $. Clearly, the ensemble size $M$ affects the (estimated) cumulative reward through term $\mathbf{E}[Z_{M_t}] $. Specifically, in the overestimation case, the estimation bias is always positive $\mathbf{E}[Z_{M_t}] >0$, and in the underestimation case, the estimation bias is always  negative $\mathbf{E}[Z_{M_t}]<0 $; the performance of the Q-learning would degrade significantly due to bias accumulation in both overestimation and underestimation cases. Thus motivated, AdaEQ adapts the ensemble size during the learning process to drive the estimation bias to be close to zero (which oscillates between positive and negative), such that the accumulated estimation bias over the learning process is minimized.
>
> Furthermore, we are able to obtain the upper bound and lower bound of $ \mathbf{E}[Q^i(s,a)]$ by bringing the results in Theorem 1 (Theorem 2). We will add the impact of the adaptive ensemble size on the cumulative reward, i.e.,  $\mathbf{E}[Q^i(s,a)] \to  y^{*} + \sum_t\alpha(1-\alpha)^t \mathbf{E}[Z_{M_t}]$,  in the form of Corollary in the revision.
>
>
> (3)[**Experiments**]
>
> Due to space limitation, we have presented the results on Humanoid-v2 in Appendix B.2 (Figure 8). Notably, given the 'non-optimal’ initial ensemble size in AdaEQ, the underestimation bias is larger than that in REDQ in the first 150 epochs.  However, the estimation bias in AdaEQ gets closer to zero after 150 epochs thanks to the ensemble size adaptation therein. The average return of AdaEQ is also better than REDQ. We will add the experimental results on Hammock and HalfCheetah in our revision.
>
> Following the existing studies on ensemble Q-learning (e.g., REDQ [6] Maxmin [16]), our experiments are carried out in the MuJoCo environment. The main reason is that Q-learning methods suffer much more from Q-function estimation bias accumulation in continuous action space compared to discrete action space [6]. This result has also been widely observed in some recent offline RL literature, e.g., ''An optimistic perspective on offline reinforcement learning'' (Agarwal et al. ICML'20). Empirically, Maxmin performs well in the discrete action space tasks but fails in most MuJoCo tasks [6]. We will add experimental results in discrete action spaces Atari (e.g., Asterix, Catcher) in our revision.

---

> > ### Comment · Reviewer_54UB · 2021-09-01
> > **Thanks for the response**
> >
> > Thanks for the clarifications. I will keep my original rating.

---

### Official Review · Reviewer_UaYC · 2021-07-18

**Rating:** 6
**Confidence:** 4

**Summary:**

This paper proposes a DRL method AdaEQ to solve the overestimation problem of Q-function. The experiment results show that AdaEQ can outperform the baseline methods.

**Limitations And Societal Impact:**

See comments in "Main Review".

**Main Review:**

The overestimation of Q is the targeting problem of this paper. The proposed AdaEQ solves this problem in a two-stage procedure: approximate error characterization and ensemble size adaptation to minimize the bias. This is a natural solution for the targeting problem.

Originality: The authors propose to optimize the estimation of Q-value through modifying the number of the ensembled Q_{n}. The authors provide the guarantees of estimation error on Q_i in the proposed method. Unlike the existing ensemble methods, AdaEQ can adaptively choose the ensemble size due to the learning process. The experiment results prove that AdaEQ works.

Clarity: The theoretical elaboration part (section3) of this paper is very detailed and logical. But the exposition of the experiment is relatively less comprehensive. The introduction of the environment and the description of the task are unclear. It is unclear whether the authors have made any adjustments and modifications to the environment of Mujoco. It would be better to add some necessary description in the experimental background. Overall, the paper is well-written and easy to understand.

Quality: The theoretical part of this article is comprehensive. The experimental part should be enhanced by adding some necessary ablation studies—for example, the effect of the choice of c (in eq.10) on the result. Similarly in eq.10, whether the different sample methods of M affect the result.

Below are some questions and suggestions for this paper.
1. In fig.5, In the Ant environment, AdaEQ has a pronounced fluctuation. Considering that Ant is a relatively stable environment, this phenomenon should not happen. How to explain this phenomenon?
2. Does AdaEQ need more compute time than REDQ? If yes, how much time does AdaEQ take more than REDQ in each environment step?
3. Is AdaEQ sensitive to c(in eq.10)?
4. How does AdaEQ perform on other Mujoco tasks? Such as humanoid, hammock? If you have done this experiment, you can also show the results, even if the results are not competitive.

Overall, the experimental results can support the claim that AdaEQ can indeed improve performance on some tasks. Although this method is straightforward, it also provides a very general idea for optimization, which is very impressive. The experimental part of this article could be enhanced to provide strong support for the claim.

=== Post Rebuttal ===

The paper proposes a novel idea with a comprehensive theoretical analysis. The experiment can provide basic support to its claims.

**Time Spent Reviewing:**

6

---

> ### Author Response · Authors · 2021-08-10
> **Reply to Reviewer UaYC**
>
> Thank you for your careful reading and constructive feedback.
>
> In terms of the  environment setting for the experiments, we use the same as in REDQ [6], AVG [2] and TD3 [8]. The detailed experiment setting is available in Appendix B. Per your suggestion, we will add more detailed description on the experimental setting in the revision.
>
> Quality: [**Sample method in Eqn. (10)**]
>
> We use random sampling in Eqn. (10) for two reasons.
>
> (i) [*Noisy approximation error*]  We use Eqn. (9) to characterize the approximation error which is noisy and time-varying in nature. In particular, Monte Carlo returns (with finite-length testing trajectory) may introduce empirical errors when estimating the underlying ground true value of $Q^{\pi}$. This estimation is noisy when the policy is not deterministic, or the environment is not deterministic. Thus, we use random sampling to take into consideration the impact of this noisy estimation.
>
> (ii) [*Impact of ensemble size on estimation bias*]	Secondly, in general it is infeasible to characterize the exact relationship between  estimation bias $Z_M$ and ensemble size $M$ (see, e.g., Section 3 in ''Toward Provable Unbiased Temporal-difference Value Estimation'', NeurIPS'19). It is this challenge that motivated us to establish the upper bound and the lower bound on the estimation bias (Theorem 1 and Theorem 2), based on which we  adapt the ensemble size, using random sampling,  in the corresponding  range (increase in $[M_t+1, N]$ or decrease in $[2,M_t-1]$) to mitigate overestimation or underestimation. Without any further prior information (except from the bounds we obtained) about the approximation error, intuitively, the random sampling can be viewed as the 'exploration' in this RL algorithm.
>
> In what follows, we provide point-to-point responses to your comments.
>
> (1) Although Ant is a relatively stable environment, the instability may often originate from the poor policy. This phenomenon has been observed in previous work, see e.g., Figure 3(e) in [6]). Intuitively, when the ensemble size is too large or too small, the estimation error accumulates quickly in the continuous action space and hence leads to a poor policy. AdaEQ is designed to overcome this issue when a 'bad’ initial ensemble size is given. In our experiments, the initial ensemble size for AdaEQ is not 'optimal' (it is obtained by grid-searching in REDQ), which yields poor performance in the early stage. However, it can be clearly observed that AdaEQ is able to drive the estimation bias close to zero during the follow-up training steps.
>
> (2) Due to space limitation, we have relegated the wall-clock training time comparison to Table 2 in Appendix B.2. It can be seen that AdaEQ need only slightly more compute time than REDQ in most MuJoCo tasks. For example, the training time of AdaEQ is 1.01x, 1.02x, 1.06x, 1.4x comparing with REDQ in Hopper, Walker, Ant, Humanoid tasks. The longer training time is partly due to the estimation bias evaluation Eqn. (9). In the experiments, we find that it suffices to adapt the ensemble size every 10 epochs so as  to reduce the estimation bias to close to zero;  this can further reduce the training time of AdaEQ. The training log is also available in the logfile folder of the supplementary files.
>
> (3) [**Ablation Study**]
>
> As shown in this work, AdaEQ is not sensitive to parameter c in the MuJoCo environment.
> (i) [*Robustness across different tasks*] In Hopper, Ant, Walker (Figure 4), Humanoid (Figure 8 in Appendix B.2), we use the same parameter $c = 0.3$ in all  MuJoCo tasks. The results shows that the same value $c=0.3$ works well to reduce the estimation error comparing with other methods.
> (ii) [*Robustness to parameter  $c$ in a wide range*]  In Figure 6(d), we compare the estimation bias over a wide range  of parameter $c$ in Hopper-v2 task. The estimation bias is consistently closer to zero when c is in range [0.3, 0.7] comparing with REDQ or AVG.
>
> In Figure 6(c), the performance of AdaEQ  degrades only when using 'extreme’ values for parameter $c$, as expected. This phenomenon is summarized in lines 271-274. In this way, AdaEQ can be viewed as a generalization of Maxmin and REDQ with ensemble size adaptation. Intuitively, when $c$ is set to be 'extreme’ large (e.g., $c=1.5$), AdaEQ  degenerates to REDQ with ensemble size 2; in contrast, when $c$ is set to be ‘extreme’ small (close to zero, e.g., $c=0.001$), AdaEQ  degenerates to Maxmin with ensemble size $N$. Both REDQ and Maxmin methods cannot  drive the estimation bias to zero during the learning process, leading to  either the overestimation or underestimation.
>
> (4) Due to space limitation, we have presented the results on Humanoid-v2 in Appendix B.2 (Figure 8). Notably, given the 'non-optimal’ initial ensemble size in AdaEQ, the underestimation bias is larger than that in REDQ in the first 150 epochs.  However, the estimation bias in AdaEQ gets closer to zero after 150 epochs thanks to the ensemble size adaptation therein. The average return of AdaEQ is also better than REDQ. We will add some experimental results on hammock in our revision.

---

> > ### Comment · Reviewer_UaYC · 2021-09-03
> > **Keep my rating.**
> >
> > Thanks for the clarifications. After reading all reviewers' comments and the authors' responses, I decided to keep my original rating.

---

### Official Review · Reviewer_hyzH · 2021-07-21

**Rating:** 7
**Confidence:** 4

**Summary:**

Some Reinforcement Learning algorithms like Q-learning are subject to a bias known as the maximization or overestimation bias. This is due to the fact that value iteration approximates the value of the next state as the max over actions of Q-values. As a consequence, if the Q-values are approximate, the maximum over such approximations tends to be skewed towards high values. Ensembling many independent estimates of the Q-values helps mitigatin the overestimation bias, but often at the risk of achieving the opposite effect, i.e. an underestimation bias, if the ensemble is too large.
The point of this paper is to propose and develop a method to estimate the current bias, and use that to estimate to dynamically choose the size of the ensemble so as to find a trade-off between over- and underestimation that minimizes the estimation bias.
The method is demonstrated on a simple toy task and then on control tasks with continuous action spaces simulated with MuJoCo physics engine.

**Limitations And Societal Impact:**

1. The paper gives a good overview of the relevant literature, but is failing to cite some recent related work like the following published papers:
- Zhu, R; Rigotti, M. 2021. Self-correcting Q-learning, AAAI 2021
- Song, Z.; Parr, R.; and Carin, L. 2019. Revisiting the Soft-max Bellman Operator: New Benefits and New Perspective. ICML 2019
- D’Eramo, C.; Nuara, A.; and Restelli, M. 2016. Estimatingthe Maximum Expected Value through Gaussian Approxi-mation. ICML 2016
It would be nice if for completeness the authors considered citing these in relation to their work.

2. The paper is proposing a new and interesting theory-driven way of mitigating the estimation bias in Q-learning. Given how general this issue is, one of the main limitations of the paper is arguably the limited number of tasks and settings under which the algorithm has been empirically evaluated. Obviously, one cannot try everything, but it would have been instructive for instance to examine at least a few tasks with discrete action spaces, like for instance Atari 2600, to verify that AdaEQ properties extend to that case and the bias estimation can be used in other settings like DQN.

3. The assumption that the approximation error is uniform across states and actions seems like a big assumption. It is true that the width of the distribution is dynamically re-estimated at each iteration, but it would be instructive if the authors had a way of showing that this is an assumption that can be borne out empirically or if they could at least elaborate on it and why it is not too strong of an assumption.

**Main Review:**

The paper starts with an illustrative synthetic example of how an approximation error in the Q-values gives rise to an estimation bias that persists in the ensemble case, and motivating that an adaptive ensemble size would actually potentially mitigate this problem.
It then develops a theory that bounds the estimation bias of an ensemble of Q-values under the assumption of an approximation error that is uniform across across states and actions, an assumption that allows the authors to postulate an analytic expression for the approximation error that is used in developing the theory. This results in analytical bounds for the bias as a function of the size of the ensemble size M and the approximation error tau.
They then go on to develop a way to estimate the approximation error tau, relying on Monte Carlo returns computed over rollouts of the current policy.
The bias mitigation algorithm AdaEQ then basically consists in using this last estimate of the approximation error tau by plugging it into the expressions for the bias in order to recover a value of the ensemble size M that results in a bias that is between the estimated upper bound indicating overestimation and lower bounds indicating underestimation.
The authors then demonstrate their algorithms against other ensemble-based bias mitigation strategies (REDQ and Average-Q) on three tasks from MuJoCO demonstrating competitive performance both in faithfully minimizing the estimation bias, and achieving higher average returns.

**Time Spent Reviewing:**

2.5

---

> ### Author Response · Authors · 2021-08-10
> **Reply to Reviewer hyzH**
>
> Thank you for your careful reading and positive feedback.
>
> (1)  We thank the reviewer for sharing the three related papers. We will cite these three papers and make comparison in our revision. Specifically, the AAAI'21 paper introduces a self-correcting estimator aiming to reduce the estimation bias, but the bias would be either positive or negative, whereas AdaEQ in this work aims to mitigate both positive and negative biases. The ICML'16 paper proposes a new estimator based on the weighted average of the sample means and conducts the empirical analysis in the discrete action space, whereas AdaEQ  uses the ensemble of Q-function approximators to control the estimation bias, guided by theoretical analysis. The ICML'19 paper suggests that the softmax operator can be used as an alternative of double Q-learning to further reduce the overestimation bias, whereas AdaEQ  uses the max operator together with a min operator over ensembles.
>
> (2) Following the existing studies on ensemble Q-learning (e.g., REDQ [6] Maxmin [16]), our experiments are carried out in the MuJoCo environment. One main reason is that Q-learning methods suffer much more from Q-function estimation bias accumulation in the continuous action space than in the discrete action space. For instance, Maxmin [16] performs well in the discrete action space tasks but fails in many MuJoCo tasks [6]. This phenomenon has also been widely observed in some recent offline RL literature, e.g., ''An optimistic perspective on offline reinforcement learning'' (Agarwal et al. ICML'20).  Evaluating AdaEQ’s performance in the discrete action space (such as Atari) is of interest, and we will include further experimental results on Atari (e.g., Asterix, Catcher) in our revision.
>
> (3) The uniform distribution assumption for the Q-function approximation error has been widely used in the existing studies (see, e.g., very recent work REDQ [6], Maxmin [16]), and this is one main reason we impose this assumption in this paper. The uniform distribution assumption for the Q-function approximation error can be traced back to Thrun et al.'s work in 1993 [27] and has also been used in the following recent papers:
>
>  - Van Hasselt et al. Deep reinforcement learning with double q-learning. AAAI'16 (where in Appendix Theorem 2, they prove an upper bound of estimation bias under the uniform distribution assumption)
>
>  - Anschel et al. Averaged-DQN: Variance reduction and stabilization for deep reinforcement learning.  ICML'17
>
>  - Wu et al. Reducing estimation bias via triplet-average deep deterministic policy gradient. IEEE TNNLS’20
>
>  - Lee et al. Choice of approximator and design of penalty function for an approximate dynamic programming based control approach. IFAC’06
>
> In fact, modeling approximation error has been an open issue, because the optimal Q-function is generally unknown and the approximation error for different tasks varies. Without any  a priori information about the approximation error, It makes sense to use a uniform distribution to indicate that both positive and negative approximation errors are equally likely in Q-function approximators.
>
> Further, we would like to point out that from a theoretic perspective, we can use the powerful machinery of Order Statistics to characterize the relationship between ensemble size $M$ and estimation bias $\mathbf{E}[Z_M]$ under general distributions of the approximation errors, but explicit bounds on the estimation bias are not attainable. Specifically, in the general case studied in Theorem 2, suppose that the approximation error $e_i(s,a)$ for each Q approximator $Q^i(s,a), i=1,2,\cdots,M$ is with PDF $f_i(x)$ and  CDF $F_i(x)$. From order statistics (see Lemma 1 in  Appendix A.1), we have the PDF and CDF for $X_{1:M} = \min_{i} e_i$ as follows:
>
> $$ F_{1:M}(x) =\mathbf{P} (X_{1:M} \leq x) = 1- \prod_{i=1}^{M}(1-F_i(x)),$$
> $$ f_{1:M}(x) =\sum_{i=1}^M \left( f_i(x)\prod_{j\neq i}\left(1-F_j(x)\right) \right),$$
>
> When there are $A$ actions applicable, the estimation bias $Z_{M}$ can be characterized by
> $$
> \mathbf{E}[Z_{M}]
> = \gamma \mathbf{E}[\max_{a'} \min_{i\in \mathcal{M}} e^i(s,a)]
> =\gamma\left[ \int_{lower}^{upper} Axf_{1:M}(x)F_{1:M}(x)^{A-1}dx \right].
> $$
>
> As expected, it is highly non-trivial, if not impossible, to derive the explicit bounds on the estimation bias when the CDF and PDF is given  in the general form.
>
> In a nutshell, in deriving Theorem 1 and 2, we have followed the the existing literature to impose the uniform distribution assumption for the Q-function approximation error . To the best of our knowledge, our work is the first to deliver both bounds under this assumption. The full proof can be found in Appendix A.1 and A.3.

---

> > ### Comment · Reviewer_hyzH · 2021-08-31
> > **Thank you for addressing my comments**
> >
> > Thank you very much for convincingly addressing all my concerns and comments.

---

### Public Comment · ~Haque_Ishfaq1 · 2022-04-21
**Code repo is empty**

Dear authors,

the provided code repo is empty. Would you consider updating it?

Thanks,

---

> ### Public Comment · Authors · 2022-04-21
> **Thanks for the request**
>
> Hi Haque, Thank you for your request. We just uploaded the code in our repo.

---

### Decision · Program_Chairs · 2021-09-27

**Decision:**

Accept (Poster)

**Comment:**

The paper presents a novel method to tackle overestimation bias in Q-learning. The majority of reviewers agreed that the paper is a good contribution to the NeurIPS community, albeit empirical evidence could be expanded further to consider additional ablation studies and environments.